# THE FALSE PROMISE OF ZERO-SHOT SUPER-RESOLUTION IN MACHINE-LEARNED OPERATORS

**Mansi Sakarvadia**[1]*, **Kareem Hegazy**[5,6], **Amin Totounferoush**[3],
**Kyle Chard**[1], **Yaoqing Yang**[4], **Ian Foster**[1], **Michael W. Mahoney**[2,5,6]
[1]University of Chicago, [2]Lawrence Berkeley National Laboratory, [3]University of Stuttgart,
[4]Dartmouth College, [5]International Computer Science Institute, [6]University of California, Berkeley

## ABSTRACT

A core challenge in scientific machine learning, and scientific computing more generally, is modeling continuous phenomena which (in practice) are represented discretely. Machine-learned operators (MLO) have been introduced as a means to achieve this modeling goal, as this class of architecture can perform inference at arbitrary resolution. In this work, we evaluate whether this architectural innovation is sufficient to perform "zero-shot super-resolution," namely to enable a model to serve inference on higher-resolution data than that on which it was originally trained. We comprehensively evaluate both zero-shot sub-resolution *and* super-resolution (i.e., multi-resolution) inference in MLOs. We decouple multi-resolution inference into two key behaviors: 1) extrapolation to varying frequency information; and 2) interpolating across varying resolutions. We empirically demonstrate that MLOs fail to do both of these tasks in a zero-shot manner. Consequently, we find MLOs are *not* able to perform accurate inference at resolutions different from those on which they were trained, and instead they are brittle and susceptible to aliasing. To address these failure modes, we propose a simple, computationally-efficient, and data-driven multi-resolution training protocol that overcomes aliasing and that provides robust multi-resolution generalization.

## 1 INTRODUCTION

Modeling physical systems governed by partial differential equations (PDEs) is critical to many computational science workflows:

$$S_2 = M(S_1), \tag{1}$$

where $M$ is an approximation of the PDE's solution operator, $S_1$ is the input state of the system, and $S_2$ is the predicted state. Central to this problem formulation is that *continuous* physical systems must be sampled and, therefore, modeled *discretely*. For a discrete model, $M$, to be useful in representing phenomena of different scales, scientists require the ability to use it at different resolutions accurately. For example, when modeling fluid flow, scientists often use adaptive mesh refinement (Berger & Oliger, 1984), a technique that increases simulation resolution in areas that require high accuracy (e.g., regions of turbulence), and coarsens it in less critical regions.

Traditionally, the approximation $M$ is constructed by numerical methods which, by design, can be employed at arbitrary discretization (Forrester et al., 2008; Cozad et al., 2014; Asher et al., 2015; Sudret et al., 2017; Alizadeh et al., 2020; Kudela & Matousek, 2022). However, numerical methods are computationally expensive. Alternatively, machine-learned operators (MLOs), a class of data-driven machine learning (ML) models which parameterize the solution operator to families of PDEs, have been proposed (Raissi et al., 2019; Li et al., 2020a; Lu et al., 2021b; Kovachki et al., 2023; Raonic et al., 2023). Although querying MLOs at arbitrary discretization is computationally inexpensive, it is not obvious that this can be done *accurately*. The Fourier Neural Operator (FNO) (Li et al., 2020a), a specific MLO, claimed to address the discretization challenge in a *zero-shot* manner (Li et al., 2020a; Tran et al., 2021; George et al., 2024; Li et al., 2024b; Azizzadenesheli et al., 2024). The claim is that FNO can be trained at resolution $m$ and then serve accurate inference at resolution

---

*Correspondence to sakarvadia@uchicago.edu, Code: https://github.com/msakarvadia/operator_aliasing

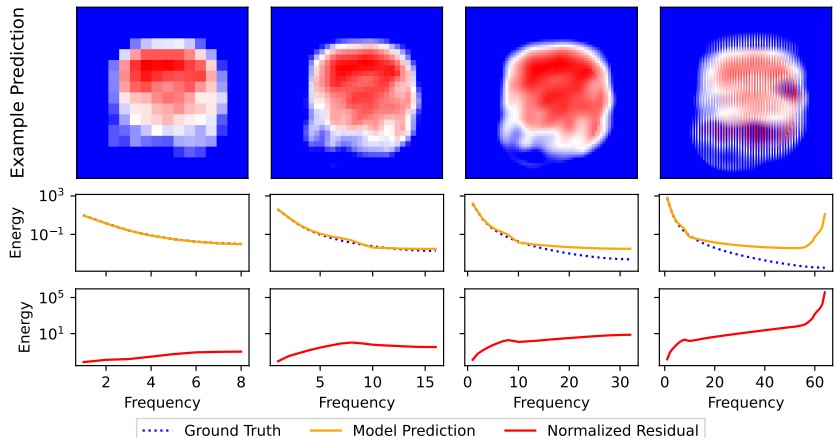

Figure 1: **Aliasing in zero-shot super-resolution.** Model trained on resolution 16 data, and evaluated at varying resolutions: 16, 32, 64, 128. **Top Row:** Sample prediction for Darcy flow; notice striation artifacts at resolution 128. **Middle Row:** Average test set 2D energy spectrum of label and model prediction. **Bottom Row:** Average residual spectrum normalized by label spectrum.

$n > m$, without training on additional high resolution data e.g., zero-shot super-resolution. This claim of zero-shot super-resolution, if true, is especially attractive in settings where generating, and training on, high-resolution data is computationally expensive.

In this paper, we evaluate the claim of zero-shot super- (and sub-)resolution inference in MLOs. We document a substantial disparity in model performance across data of different discretizations, suggesting that MLOs are generally incapable of accurate inference at resolutions greater or less than their training resolution (i.e., zero-shot multi-resolution inference). Instead, we find that MLOs often misrepresent unseen frequencies and incorrectly infer their behavior in data whose discretization differs from its training discretization, i.e., they exhibit a form of aliasing (Fig. 1). In addition, we study two previously proposed solutions: *(i)* physics-informed optimization constraints (Li et al., 2024b) and *(ii)* band-limited learning (Raonic et al., 2023; Gao et al., 2025). We find that neither enables zero-shot multi-resolution, as they do *not* address the central issue: MLOs, like all machine-learned models, cannot typically generalize beyond their training data (Yang et al., 2023; Liu et al., 2020; Krueger et al., 2021). We establish that the discretization at which MLOs are trained impacts the discretization at which they accurately model the system.

To enable multi-resolution inference, we propose *multi-resolution training*, a simple, intuitive, and principled data-driven approach which trains models on data of multiple resolutions. We profile the impact of different multi-resolution training approaches, finding that optimal multi-resolution performance can often be achieved via training data sets that contain mostly low resolution (less expensive) data and very little high resolution (more expensive) data. This permits us to achieve low computational overhead, while also increasing the utility of a single MLO.

To summarize, the main contributions of our work are the following:

1. We assess the ability of trained MLOs to generalize beyond their training resolution. We demonstrate that MLOs struggle to perform accurate inference at resolutions higher or lower than which are they trained on, and instead they exhibit aliasing. Based on these results, we conclude that accurate *zero-shot* multi-resolution inference is unreliable (Sec. 3).

2. We evaluate two intuitive approaches—incorporating physics-informed constraints during training, as well as performing band-limited learning—and we find that neither approach enables reliable multi-resolution generalization. (Sec. 4).

3. We propose and test multi-resolution training, where we include training data of varying resolutions (in particular, a small amount of expensive higher-resolution data and a larger amount of cheaper lower-resolution data), and we show that multi-resolution inference improves substantially, without a significant increase in training cost (Sec. 5).

| (a) Original | (b) Interpolate | (c) Extrapolate | (d) Super-Resolution | (e) Alias |
|---|---|---|---|---|
| Signal Freq.=3
Sample Freq.=40 | Signal Freq.=3
Sample Freq.=70 | Signal Freq.=7
Sample Freq.=40 | Signal Freq.=7
Sample Freq.=70 | Signal Freq.=7
Sample Freq.=8 |

— ● — Sampling Rate      ⋯⋯ True signal

Figure 2: **Accurate multi-resolution inference requires both interpolation & extrapolation.** **Original:** Signal is sampled at a rate greater than its Nyquist frequency. **Interpolation:** Adapting to new sampling rates of a given signal. **Extrapolation:** Adapting to new frequency information under constant sampling rate. **Super-Resolution:** Sampling a system at a higher rate which enables the capture of higher frequency information (interpolation & extrapolation). **Aliasing:** High-frequency information is misrepresented as a low-frequency information due to insufficient sampling.

## 2 BACKGROUND ON SIGNAL PROCESSING AND ALIASING

We start by discussing the practice of training ML models to represent continuous systems via discrete data. Next, we outline the implications of aliasing in ML as it relates to multi-resolution inference. Finally, we formally define "zero shot multi-resolution" inference in a discrete context.

**Discrete Representations of Continuous Systems.** The fundamental challenge in discretely representing continuous systems lies in the choice of sampling rate. The Whittaker–Nyquist–Shannon sampling theorem established that given a sampling rate $r$, the largest resolved frequency is $r/2$ (Unser, 2002; Shannon, 1949; Whittaker, 1928). Thus, ML models will be trained on discrete representations where only some of the frequencies are fully resolved. Resolving higher-order frequencies greater than $r/2$, consequently, becomes an out-of-distribution task. Aligning these discrete models' predictions with the underlying continuous system is an open problem (Krishnapriyan et al., 2021; Queiruga et al., 2020; Ott et al., 2021; Ren et al., 2023; Takamoto et al., 2022; Subel et al., 2022; Chattopadhyay et al., 2024).

**Aliasing.** When sampling a continuous signal at rate $r$, aliasing occurs when frequency components greater than $r/2$ are projected onto lower frequency basis functions (Fig. 2) (Gruver et al., 2022). Thus, content with frequency $n > r/2$ is observed as a lower frequency contribution:

$$\text{Alias}(n) = \begin{cases} n \bmod r & \text{if } (n \bmod r) < r/2 \\ r - (n \bmod r) & \text{if } (n \bmod r) > r/2 \end{cases} \tag{2}$$

In an ML context, when inferring at different discretizations of a given signal, aliasing can manifest as the divergence between the energy spectrum of the model prediction and the expected output. Aliasing indicates a model's failure to fit the underlying continuous system.

**Zero-shot multi-resolution inference.** We define *multi-resolution* inference as the ability to do inference at multiple resolutions (e.g., sub- and super-resolution). The *zero-shot* multi-resolution task employs an ML model, which is trained on data with some resolution and tested on data with a different resolution. Zero-shot multi-resolution inference raises two important questions with respect to the generalization abilities of trained models (see Fig. 2):

1. **Resolution Interpolation.** How do models behave when the frequency information in the data remains fixed, but its sampling rate changes from training to inference? Can the model *interpolate* the fully resolved signal to varying resolutions?

2. **Information Extrapolation.** How do models behave when the resolution remains fixed, but the number of fully resolved frequency components changes from training to inference? For super-resolution, this means can the model *extrapolate* beyond the frequencies in its training data and model higher frequency information?

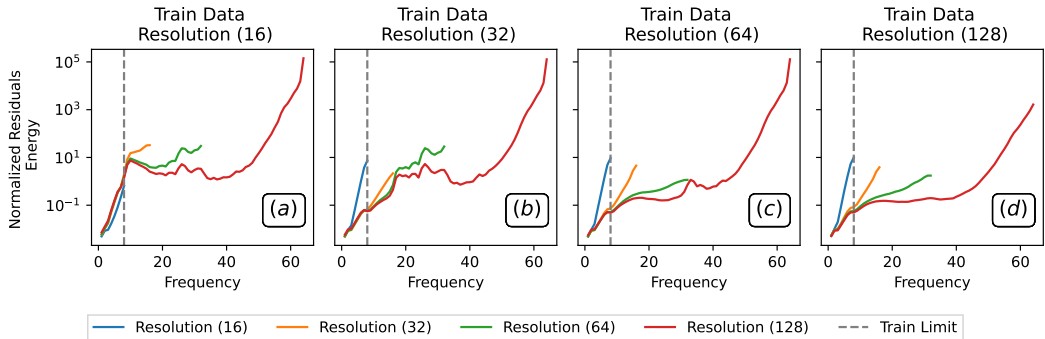

Figure 3: **Resolution Interpolation.** Four FNOs are trained on Darcy data at resolutions $\{16, 32, 64, 128\}$ from left to right with constant frequency information (low-pass limit of $8f$), and are tested on varying resolutions. We assess if each model can generalize to data with varying sampling rate. We visualize spectra of the normalized residuals across test data. Notice, residual spectra (error) increases substantially in the low frequencies. Lower residual energy at all frequencies is better.

## 3 ASSESSING MULTI-RESOLUTION GENERALIZATION

We examine the zero-shot multi-resolution abilities of FNO, an architecture for which this claim has been previously made (Li et al., 2020a). We study the multi-resolution inference task from an out-of-distribution data perspective by decoupling what it means for a model to perform inference at a resolution different from that used during training. Specifically, in Sec. 3.1, we assess whether models trained on a system sampled at rate $r$ are capable of both interpolating accurately to **new** sampling rates (Fig. 2(b)) *and* extrapolating accurately to **additional/fewer** frequencies present in data (Fig. 2(c)). We systematically test an FNO's ability to do both objectives and show neither are achieved. In light of these failure modes, in Sec. 3.2, we then assess the spatial zero-shot sub- and super-resolution capabilities of FNOs and show the claim does not hold (Fig. 2(d)).

We evaluate FNO on three standard scientific datasets: Darcy, Burgers, and Navier Stokes. For each dataset we optimize FNO hyperparameters via grid-search as described in Appendix B.

### 3.1 BREAKING DOWN MULTI-RESOLUTION CAPABILITIES

**Resolution Interpolation.** We assess if an FNO trained on data of a specific resolution can generalize to data of both lower and higher resolution under fixed frequency information. Specifically, we keep the set of populated frequencies constant in the train and test data, while varying the resolution of the test data. We do this by applying a low-pass filter to all data at the highest resolution, and then down-sampling as needed. The sampling rates of all data are large enough to resolve all remaining frequencies.

We begin with a simple experiment: we train an FNO on a Darcy flow dataset at resolution $N = 16$ and assess the trained model's performance across test datasets at varying resolutions $\{16, 32, 64, 128\}$, all low-pass filtered with limit $8f$ where $f$ is the frequency unit $2\pi/N$. In Fig. 3(a), we visualize the average spectral energy of the model predictions normalized by the spectral energy of the unfiltered ground truth for each test dataset. For the test data with resolutions that are different from the training data, we observe sharp increases in their residual's energy spectrum in frequencies greater than $8f$. This is especially concerning since the model was never trained on nor shown inference data containing frequencies greater than $8f$. In other words, FNOs, trained in a zero-shot manner, fail to reliably *interpolate* to varying resolutions.

In Figs. 3(b-d), we repeat the same experiment at varying training resolutions (e.g, 32, 64, 128) with low-pass limit $8f$. We notice that at each training resolution, the model consistently and incorrectly assigns high energy in frequencies greater than $8f$ across all test resolutions. We perform corresponding experiments for the Burgers dataset with low-pass limit $64f$ and resolutions $\in \{128, 256, 512, 1024\}$ and Navier Stokes dataset with low-pass limit $32f$ and resolutions $\in \{64, 128, 255, 510\}$ and observe the same failure mode (Appendix C: Figs. 11-12). **We conclude that changing**

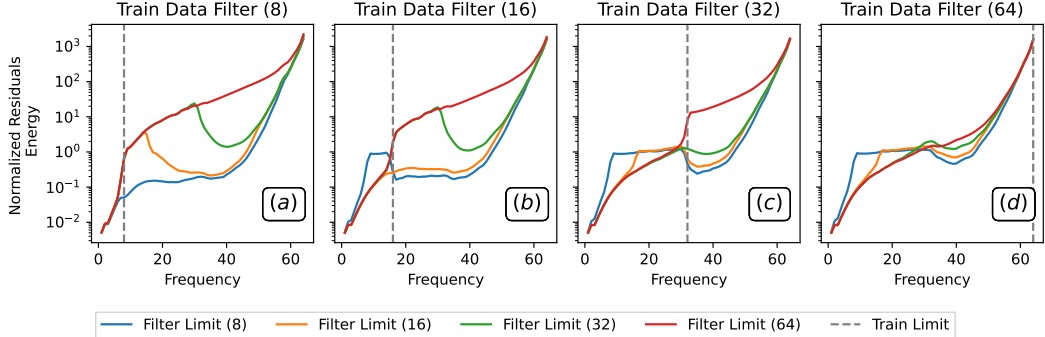

Figure 4: **Information Extrapolation.** Four FNOs trained on Darcy data of resolution 128 (constant sampling rate) and low-pass filtered with limits $\{8, 16, 32, 64\}f$ (varying frequency information) from left to right. Test if each model can generalize to data with varying frequency information. Visualizing spectrum of the normalized residuals across test data. Notice, residual spectra (error) increases substantially in the high frequencies. Lower residual energy at all frequencies is better.

**resolution at test time is akin to out-of-distribution inference:** the model is never trained on data with a broad range of sampling rates and consequently fails to generalize.

**Information Extrapolation.** We assess if an FNO trained on data containing a fixed set of frequencies can generalize to data containing both fewer and additional frequencies under fixed resolution. Specifically, we keep the resolution constant but vary the number of populated frequencies in the train and test datasets by applying varying low-pass filters to data at a fixed resolution.

We begin with a simple experiment: an FNO is trained on a Darcy flow dataset at resolution 128 which is low-pass filtered at limit $8f$. In Fig. 4($a$), we assess the trained model's performance across four versions of a test dataset, all of which have the same sampling resolution (128) but are filtered with low-pass limits $\{8, 16, 32, 64\}f$ (e.g., increasing amounts of frequency information). There is a sharp increase in the residual spectra in higher frequencies across all test sets; the residual error increases as the test and training filters diverge. In other words, FNOs, trained in a zero-shot manner, fail to *extrapolate* on data with previously unseen frequency information.

In Figs. 4($b$-$d$), we repeat the same experiment at varying training data low-pass filter limits (e.g, 16, 32, 64$f$). Each model consistently and incorrectly assigns high energy in the high frequencies regardless of whether the test data contained any high-frequency information. This is a concerning failure mode indicating FNOs do not generalize both in the presence of frequencies greater than, *and* the absence of frequencies less than, what was present in their training data. We perform corresponding experiments for the Burgers dataset with resolution 1024 and low-pass limits $\in \{64, 128, 256, 512\}f$ and Navier Stokes dataset with resolution 510 and low-pass limits $\in \{8, 16, 32, 255\}f$ and observe the same failure mode (Appendix C: Figs. 13-14). **We conclude that *varying* the frequency information at test time is effectively out-of-distribution inference** as the model was not trained on data with such variation in frequency information and, therefore, fails to generalize.

### 3.2 ZERO-SHOT MULTI-RESOLUTION INFERENCE

We study whether FNOs are capable of spatial multi-resolution inference: simultaneously changing the sampling rate and frequency information. For each dataset in {Darcy, Burgers, Navier Stokes}, we train a model on data at resolutions (16, 32, 64, 128), (1024, 512, 256, 128,), (510, 255, 128, 64), respectively. In Fig. 1, we see that models trained at low resolutions do not generalize to high resolutions. Similarly, in Fig. 5, we again see a failure to generalize and instead observe high-frequency artifacts in model predictions in multi-resolution settings. Further, for time-varying PDEs, such as Navier Stokes, we observe that these high frequency aliasing artifacts compound across time steps (Fig. 15). In Fig. 16, we show that models trained at a given resolution do not achieve low loss across all test resolutions. In fact, losses vary by $1\times$, $2\times$, and $10\times$ across test resolutions for the Darcy, Burgers, and Navier Stokes datasets, respectively. **Therefore, we conclude that Fourier neural operators are not capable of consistent *zero-shot* super- or sub-resolution.**

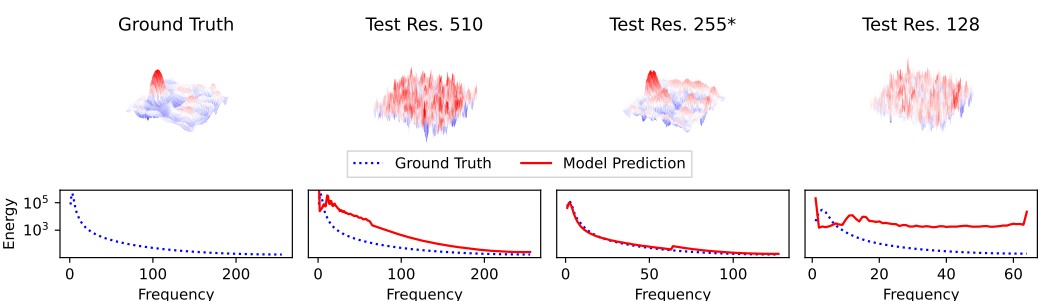

Figure 5: **FNOs do not generalize to higher or lower resolutions.** Model trained on Navier Stokes dataset at resolution 255 (indicated by *), evaluated resolutions 510, 255, 128. **Top Row:** Ground truth, prediction at resolutions 510 (super-resolution), resolution 255 (same as train resolution), and resolution 128 (sub-resolution). **Bottom Row:** Average energy spectra over test data.

## 4    EVALUATING POTENTIAL CORRECTIVE METHODS

Here we assess two previously proposed strategies for accurate zero-shot multi-resolution inference: physics-informed optimization objectives (Li et al., 2024b) and band-limited learning (Raonic et al., 2023; Gao et al., 2025). We find that neither enables accurate multi-resolution inference.

### 4.1    PHYSICAL OPTIMIZATION CONSTRAINTS

Physics-informed optimization constraints have been proposed as a means of achieving accurate inference in the *zero-shot* super-resolution setting (Li et al., 2024b). For each dataset in {Darcy, Burgers, Navier Stokes}, we train FNOs at all avaliable resolutions. We optimize each set of model parameters $\theta$ with a dual optimization objective $\mathcal{L}(\theta) = (1 - w) * \ell_{\text{data}}(\theta) + w * \ell_{\text{phys}}(\theta)$, where $\ell_{\text{data}}$ is the original data-driven loss (mean squared error) with an additional physics-informed loss $\ell_{\text{phys}}$, which explicitly enforces that the governing partial differential equation is satisfied. We use the physics losses of Li et al. (2024b) and detail the implementation in Appendix E.

We find that the data-driven loss always outperforms any training objective that includes a physics constraint (Fig. 17). We determine this by evaluating $w \in \{0, 0.1, 0.25, 0.5\}$ for {Darcy, Burgers, Navier Stokes} at resolutions 64, 512, 255, respectively. Among the objectives that contain a physics constraint, we observe a clear trend: the lower the physics constraint is weighted, the better test performance the model achieves. This indicates that physics constraints make it more challenging for the model to be trained optimally despite extensive hyperparameter tuning (perhaps due to practical reasons such as being difficult to optimize (Krishnapriyan et al., 2021; Subramanian et al., 2022; Gao & Wang, 2023; Wang et al., 2023)).

To further illustrate, we use the smallest $w = 0.1$ and compare the physics-informed optimization with solely data-driven optimization. In Fig. 6, the predicted spectra of data from models optimized with physics loss generally diverge more substantially across test resolutions than models optimized with only a data loss. Models optimized with physics constraints even fail to accurately fit their training distributions (Fig. 6(c)), and fail to generalize to both super- and sub-resolution data (Fig. 6(a,b,d)). See Appendix E for results on all datasets. We conclude that physics informed constraints do *not* reliably enable multi-resolution generalization.

### 4.2    BAND-LIMITED LEARNING

We study two approaches which propose learning band-limited representations of data: convolutional neural operators (CNO) (Raonic et al., 2023) and the Cross-Resolution Operator-Learning (CROP) pipeline (Gao et al., 2025). Both CNO and CROP have been proposed as alias-free solutions to enable multi-resolution inference (Bartolucci et al., 2023). CNOs are *fixed-resolution* models; to use them, one must first interpolate their input to the model's training resolution, do a forward pass, and then interpolate back to the original dimension. The CROP pipeline is more gen-

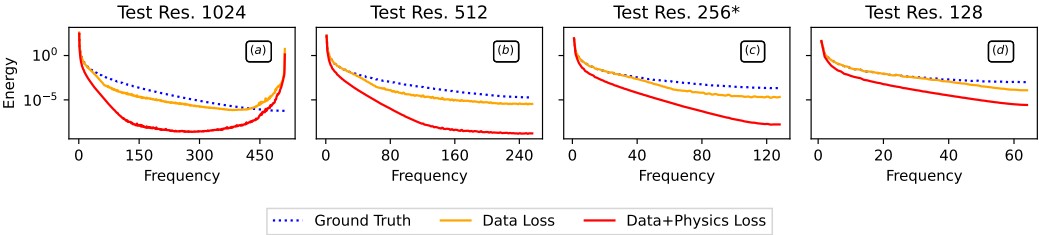

Figure 6: **(Physics-Informed) Optimization Evaluation.** FNO trained on Burgers data at resolution 256 (indicated by *) with and without physics optimization constrains. Average test spectra visualized. Spectra generated by the models trained with physics+data loss do not match the ground truth. Physics Loss term weighting $w = 10\%$. Full results in Appendix E, Figs. 19-21.

eral and extends to any class of model by interpolating inputs (both at training and inference) to and from a fixed-dimensional representation before and after a forward pass.

We train and evaluate a CNO and CROP+FNO model on all {Darcy, Burgers, Navier Stokes} datasets. Models are optimized using optimal hyper-parameters which were found via a grid search detailed in Appendix B. In Fig. 7, we visualize the predicted spectra of model's trained on resolution 16 data across test data of resolution {16, 32, 64, 128}. We observe that the CNO does accurately learn the band-limited representation of its training data: the spectra matches that of the ground truth until frequency $8f$ after which, by design, it drops sharply. This means that while CNO does not alias, it cannot predict frequencies higher than what was seen during training. Similarly, we observe that the CROP model, accurately fits lower frequencies, but struggles to fit high frequencies accurately across resolutions.

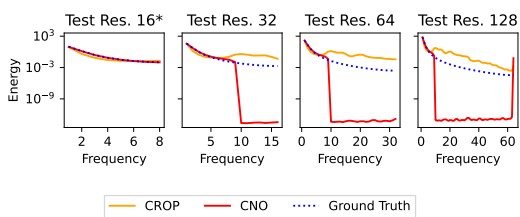

Figure 7: **Bandlimited Learning Evaluation.** Models trained on Darcy dataset at resolution 16 (indicated by *), evaluated resolutions 16, 32, 64, 128. Average test set 2D energy spectrum of model predictions and ground truth. Notice both model prediction spectrums diverge from ground truth after frequency 8.

We note more broadly that the design of band-limiting a model's training data and predictive capacity is counter-intuitive to the goal of multi-resolution inference, in which, a broad range of frequencies must be modeled accurately. **Band-limiting a model's predictive capacity may enable accurate fixed-resolution representations, but ensures that high-frequency information is not predicted accurately (or at all).** We conclude that band-limited learning limits a model's utility for *multi-resolution* inference (full results in Appendix F).

## 5 MULTI-RESOLUTION TRAINING

We hypothesize that the reason models struggle to do *zero-shot* multi-resolution inference is because data representing a physical system at varying resolutions is sufficiently *out-of-distribution* to a model's fixed-resolution training data. To remedy this, we propose a data-driven solution: multi-resolution training (i.e., training on more than one resolution) (Li et al., 2024a; Rowbottom et al., 2025; Berner et al., 2025; George et al., 2024; Tang et al., 2024; Lyu et al., 2023; Ma et al., 2024; Lanthaler et al., 2024).

We compose multi-resolution datasets by randomly sampling different proportions of training data at varying resolutions $\{r_1, ..., r_n\}$ where $r_x$ is the proportion of $x$ resolution training data and $n = 4$. We begin by evaluating dual-resolution training. Li et al. (2024a) have previously shown that dual-resolution active learning enables more accurate *high-resolution* inference. We extend this finding and evaluate if dual-resolution training can enable accurate *multi-resolution* inference. For each dataset in {Darcy, Burgers, Navier Stokes}, we combine data across resolutions in a pairwise manner

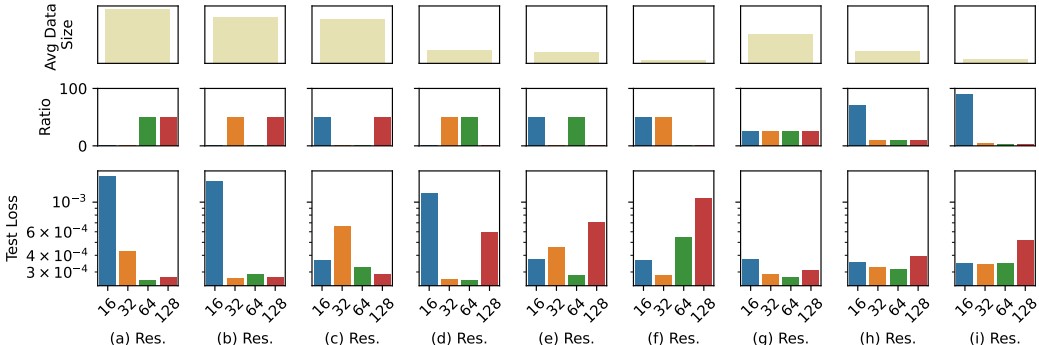

Figure 8: **Multi-resolution training.** FNO trained on multi-resolution Darcy data. **Top row:** Average number of pixels in a data sample in the training set. Lower number of pixels enables faster data generation and model training. **Middle row:** The ratios of data within each resolution bucket. **Bottom row:** The average test loss across different resolutions. Lower loss is better. Notice: mixed resolution datasets achieve both low average data size and low loss (ideal scenario).

creating $\frac{n(n-1)}{2}$ dual-resolution sets; the ratio of data samples between the two resolutions is varied over $p \in \{0.5, 0.1, 0.25, 0.5, 0.75, 0.9, 0.1\}$. In Fig. 8(a-f), we observe for pair-wise training, the test performance for data that corresponds to the two training resolutions is generally better, but there are not consistent gains for the two non-training resolutions. This indicates that models perform best on the data resolutions on which they are trained.

To improve multi-resolution capabilities, we investigate the impact of including data from *all* resolutions. We first assess an equal number of samples across resolutions. In Fig. 8(g), the test performance across all resolutions improves which confirms that multi-resolution training benefits multi-resolution inference. Next, we ask: *Can we improve the computationally efficiency of multi-resolution training?* To do this, the training dataset must be composed of primarily low resolution data as it is both the cheapest to generate and train over (Fig. 32). We compose two additional multi-resolution datasets: $\{(0.7, 0.1, 0.1, 0.1), (0.9, 0.5, 0.3, 0.2)\}$. In Fig. 8(h, i), models remain competitive across test resolutions, even as we decrease the amount of high-resolution data. In Fig. 9, we observe the consistent trend for all datasets: models are able to achieve a balance between dataset size and multi-resolution test loss via multi-resolution training. Optimizing the ratios across all resolutions remains an exciting future direction. Full results in Appendix G.

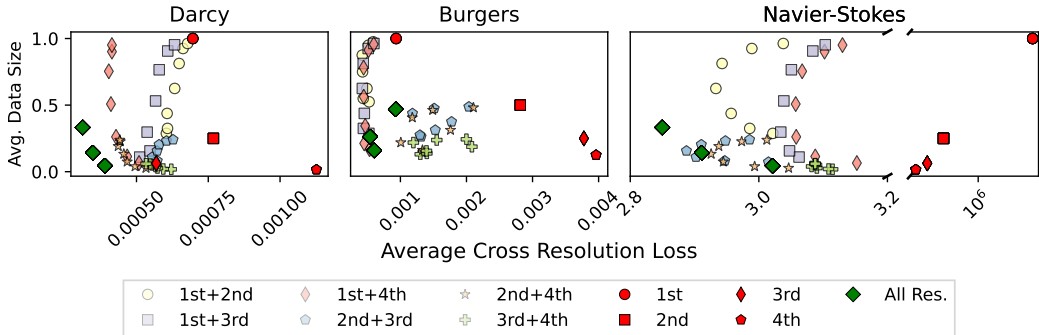

Figure 9: **Data Cost vs. Loss Tradeoff.** Lower data size and test loss are desirable (bottom left corner). We generally notice the "All Res." datasets form the Pareto front of achieving optimal data size vs. test loss. "$m$" and "$m+n$" indicate the one or two resolutions that were included in the training set. "All Res." indicates that dataset contains points from all resolutions available. "Avg. Data Size" is the normalized average number of pixels in a data point.

## 6 RELATED WORK

**Modeling PDEs via Deep Learning.** Three prominent approaches exist to discretely model PDEs with deep learning. **1) Mesh-Free** models learn the solution operator to a *specific* instance of a PDE (Yu et al., 2018; Raissi et al., 2019; Bar & Sochen, 2019; Smith et al., 2020; Pan & Duraisamy, 2020). Mesh-free models can be queried to return a measurement at any time and space coordinate. While, this approach means that a single model can resolve its solution at arbitrary discretization, it has two shortcomings: (i) Inference costs increases with number of points queried. (ii) Models cannot generalize beyond the specific PDE instance it was trained on. **2) Fixed-Mesh** models remedy both issues by learning a solution operator for a PDE *family* over a fixed-resolution mesh (Guo et al., 2016; Zhu & Zabaras, 2018; Adler & Öktem, 2017; Bhatnagar et al., 2019; Khoo et al., 2021). The inference costs of fixed-mesh models are lower than traditional numerical methods at corresponding mesh resolutions. However, fixed-mesh models fall short when one is interested in modeling scale phenomena that cannot be resolved via the fixed-mesh resolution (e.g., high-frequency information in turbulent systems ). **3) Mesh-invariant** models, unlike fixed-mesh models, are capable of doing inference at arbitrary mesh resolutions (Li et al., 2020b;a; Lu et al., 2021b; Bhattacharya et al., 2021; Nelsen & Stuart, 2021; Patel et al., 2021; Rahman et al., 2022; Fanaskov & Oseledets, 2023). They have been proposed as a means to learn **mesh-invariant** solution operators to entire PDE families cheaply: train on low-resolution data, and use in a zero-shot fashion on arbitrary resolution data (e.g., zero-shot multi-resolution). In this work, we examine the zero-shot multi-resolution utility of mesh-invariant models.

**Aliasing (and corrective measures) in Deep Learning.** Sources of aliasing in deep learning include both artifacts of a pixel grid which models learn and amplify and the application of point-wise non-nonlinearities to intermediate model representations (Karras et al., 2021; Gruver et al., 2022; Wilson, 2025). A straightforward approach, first introduced in generative adversarial networks, to stem nonlinearity-caused aliasing is to up sample a signal before applying a non-linearity followed by down sampling the signal (Karras et al., 2021). Bartolucci et al. (2023) and Raonic et al. (2023) extend the application of anti-alias activation function design to scientific machine learning; while this does prevent *aliasing*, it does not enable models trained at a specific resolution to resolve higher frequencies in higher resolution data. Gao et al. (2025) proposed a framework that "lifts" arbitrarily discretization data to a fixed-resolution band-limited space, to both train and do inference in. We investigate the sources of aliasing the context of *zero-shot* multi-resolution inference and show that proposed solutions fall short in remedying the core issue: out-of-distribution generalization.

## 7 CONCLUSION AND FUTURE WORK

For machine-learned operators to be as versatile as numerical methods-based approaches for modeling PDE's they must perform accurate multi-resolution inference. To better understand an MLO's abilities, we break down the task of multi-resolution inference and assess a trained model's ability to both *extrapolate* to higher/lower frequency information in data and *interpolate* across varying data resolutions. We find that models trained on low resolution data and used for inference on high-resolution data can neither extrapolate nor interpolate, and therefore, more generally fail to do accurate multi-resolution inference. Changing the resolution of data at inference time is akin to out-of-distribution inference: models have not learned how to generalize in such settings. We document that models used in a *zero-shot* multi-resolution setting are prone to aliasing. We study the utility of two existing solutions–physics-informed constrains and learning band-limited learning–and find that neither enable accurate multi-resolution inference.

We introduce a simple and principled approach to enable accurate multi-resolution inference: multi-resolution training. We first show that models perform best at resolutions they have been trained on. We then extend this finding and demonstrate that one can computationally efficiently achieve the benefits of multi-resolution training via datasets composed with mostly low-resolution data and small amounts of high-resolution data. This enables accurate multi-resolution learning with the added benefit of low data-generation and model training cost. A promising future direction remains the automated selection of multi-resolution training data using strategies like active learning.

## ACKNOWLEDGEMENTS

This material is based upon work supported by the U.S. Department of Energy, Office of Science, Office of Advanced Scientific Computing Research, Department of Energy Computational Science Graduate Fellowship under Award Number DE-SC0023112. YY acknowledges support from DOE Award DE-SC0025584 and Dartmouth College. AT was funded by Deutsche Forschungsgemeinschaft (DFG, German Research Foundation) under Germany's Excellence Strategy - EXC 2075 – 390740016 and acknowledges the support by the Stuttgart Center for Simulation Science (SimTech). We would also like to acknowledge the DOE Competitive Portfolios grant and the DOE SciGPT grant.

## REPRODUCIBILITY STATEMENT

To enable reproduction of our results we will link to the code base upon acceptance. We additionally include details about hyperparameter tuning in Appendix B, models (and their configuration details) in Appendix H, datasets (and their configuration details) in Appendix A, data extrapolation/interpolation implementation details in Appendix C.1. To ensure reliability of results we include an additional sensitivity analysis in Appendix L. Unless otherwise specified, all experiments were run using Python 3.10 on an NVIDIA A100-PCIE-40GB with driver version 550.163.01; CUDA Toolkit version was 12.4.131; PyTorch version was 2.9.1+cu126; the "neuraloperator" library (Kossaifi et al., 2025) version was 2.0.0. We report results from replicating a subset of our experiments across multiple computing architectures to ensure reproducibility in Appendix N.

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

# A    DATA

We study three standard scientific datasets: Darcy, Burgers, and turbulent incompressible Navier Stokes, as released in PDEBench (Takamoto et al., 2022). We summarize each dataset here; for full details, see Takamoto et al. (2022):

**Darcy.** We study the steady-state solution of 2D Darcy Flow over the unit square with viscosity term $a(x)$ as an input of the system. We learn the mapping from $a(x)$ to the steady-state solution described by:

$$-\nabla(a(x)\nabla u(x)) = f(x), \ x \in (0,1)^2$$
$$u(x) = 0, \ x \in \partial(0,1)^2.$$

The force term is a constant value $f = 1$.

**Burgers.** We study Burgers' equation which is used to model the non-linear behavior and diffusion process in fluid dynamics:

$$\partial_t u(t,x) + \partial_x(u^2(t,x)/2) = v/\pi \partial_{xx} u(t,x), \ x \in (0,1), t \in (0,2] \tag{3}$$

$$u(0,x) = u_0(x), \ x \in (0,1). \tag{4}$$

The diffusion coefficient is a constant value $f = 0.001$.

**(Turbulent) Inhomogeneous, Incompressible Navier Stokes.** We study a popular variant of the Navier Stokes equation: the incompressible version. This equation is used to model dynamics far lower than the speed of propagation of waves in the medium:

$$\nabla \cdot v = 0, \ \rho(\partial_t v + v \cdot \nabla v) = -\nabla p + \eta \Delta v. \tag{5}$$

Takamoto et al. (2022) employ an augmented form of (5) which includes a vector field forcing term $u$:

$$\rho(\partial_t v + v \cdot \nabla v) = -\nabla p + \eta \Delta v + u.$$

The viscosity is a constant value v=0.01. We convert the incompressible Navier Stokes dataset to vorticity form to enable direct comparison with Li et al. (2020a).

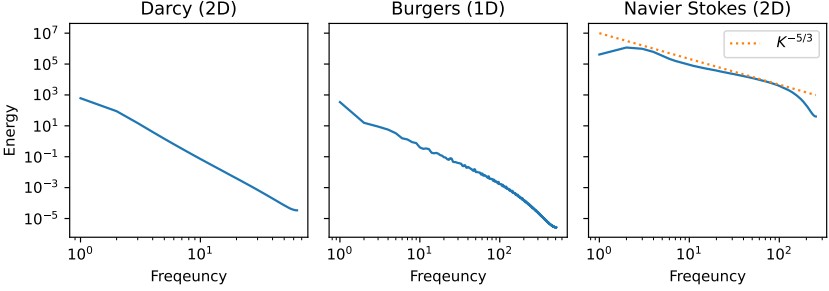

Figure 10: **Dataset Energy Spectrum.** Average energy spectra over test datasets. Notice that Navier Stokes is in the turbulent regime. K = Kolmogorov coefficient.

# B HYPER-PARAMETER SEARCH

**FNO Hyper-parameter Tuning.** For each dataset $\in$ {Darcy, Burgers, Navier Stokes}, we do a grid search for optimal training hyper-parameters: learning rate $\in$ {1e-2, 1e-3, 1e-4, 1e-5}; and weight decay $\in$ {1e-5, 1e-6, 1e-7}; for both the data driven loss (i.e., mean squared error) and the respective data+physics driven loss. Each model was trained for 150 epochs. For the models optimized with the data+physics loss, we optimized the physics loss term's weighting coefficient $w \in$ {0.1, 0.25, 0.5}. For Darcy, Burgers, and Navier Stokes, we do this hyper-parameter search for data at resolution 64, 512, 255, respectively, and we then use the optimized parameter values for each dataset to train models at the remaining resolutions. See Tab. 1 for the optimal hyper-parameter for each dataset/loss combination.

**CROP/CNO Hyper-parameter Tuning.** For each dataset $\in$ {Darcy, Burgers, Navier Stokes}, we do a grid search for optimal training hyper-parameters: learning rate $\in$ {1e-3, 1e-4, 1e-5}, for the data driven loss (i.e., mean squared error). Each model was trained for 150 epochs. For Darcy, Burgers, and Navier Stokes, we do this hyper-parameter search for data at resolution 64, 512, 255, respectively; and we then use the optimize parameter values for each dataset to train models at remaining resolutions. See Tab. 2 for the optimal hyper-parameter for each dataset/loss combination.

Table 1: **Optimal FNO hyper-parameters** from hyper-parameter search outlined in Appendix B. *NS batch size had to be reduced to 1 for multi-resolution training experiments (see Sec. 5), therefore we used a lower learning rate in that setting. $w$=Physic Loss Coefficient (see Sec. 4.1).

| Data | Loss | $w$ | Learning Rate | Weight Decay | Batch Size |
|------|------|-----|---------------|--------------|------------|
| Darcy | Data | - | 1e-3 | 1e-5 | 128 |
| Darcy | Data+Physics | 0.1 | 1e-2 | 1e-5 | 128 |
| Burgers | Data | - | 1e-3 | 1e-5 | 64 |
| Burgers | Data+Physics | 0.1 | 1e-3 | 1e-5 | 64 |
| Navier Stokes | Data | - | 1e-2 | 1e-6 | 4 |
| Navier Stokes | Data+Physics | 0.1 | 1e-4 | 1e-5 | 4 |
| Navier Stokes* | Data | - | 1e-5 | 1e-6 | 1 |

Table 2: **Optimal CNO/CROP/DeepONet hyper-parameters** from hyper-parameter search outlined in Appendix B. *The original CROP implementation did not include a 1D version, so we omit CROP for the 1D Burgers dataset.

| Data | Loss | Model | Learning Rate | Weight Decay | Batch Size |
|------|------|-------|---------------|--------------|------------|
| Darcy | Data | CNO | 0.0001 | 1e-5 | 128 |
| Darcy | Data | CROP | 0.001 | 1e-5 | 128 |
| Darcy | Data | DeepONet | 0.001 | 1e-5 | 128 |
| Burgers* | Data | CNO | 0.001 | 1e-5 | 64 |
| Navier Stokes | Data | CNO | 0.001 | 1e-6 | 1 |
| Navier Stokes | Data | CROP | 0.001 | 1e-6 | 1 |

## C  INFORMATION EXTRAPOLATION AND RESOLUTION INTERPOLATION

### C.1  EXTRAPOLATION AND INTERPOLATION IMPLEMENTATION DETAILS

**Down sampling Procedure:** For the resolution interpolation experiments, data was sampled at different resolution by employing standard down sampling of the data (keeping every Nth element) from the maximal data resolution. This is the same down sampling procedure used by PDEBench (Takamoto et al., 2022).

**Low-pass Filtering Procedure:** For the information extrapolation experiments, data was filtered with a standard stop-band filter which restricts any frequency components above the desired filter limit $N$. Concretely, to low pass filter data, we transform the data to the frequency domain via an FFT and shift it such that the lowest frequency component is the center of the spectrum. We determine which frequency components are greater than our limit $N$ by assessing which components fall outside a radius of $N$ from the center; we then set the energy of any frequency components greater than $N$ to zero. Finally, the (filtered) data is transformed back to the spatial domain via an inverse FFT.

### C.2  FULL RESULTS

Here, we include the full experimental results of studying FNOs' abilities to do both information extrapolation and resolution interpolations, as described in Sec. 3.

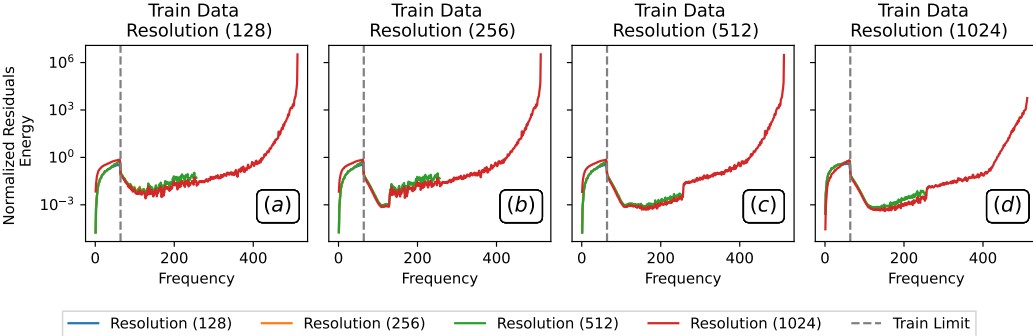

Figure 11: **Interpolation.** Four FNOs trained on Burgers data low-pass limit $64f$ (constant frequency information) and down sampled to resolutions $\{128,256,512,1024\}$ (varying sampling rate) from left to right. We test if each model can generalize to data with varying sampling rate. We visualize the normalize residual spectra across test data. Notice that the residual spectra (error) increases substantially in the low frequencies. Lower energy at all wave numbers is better.

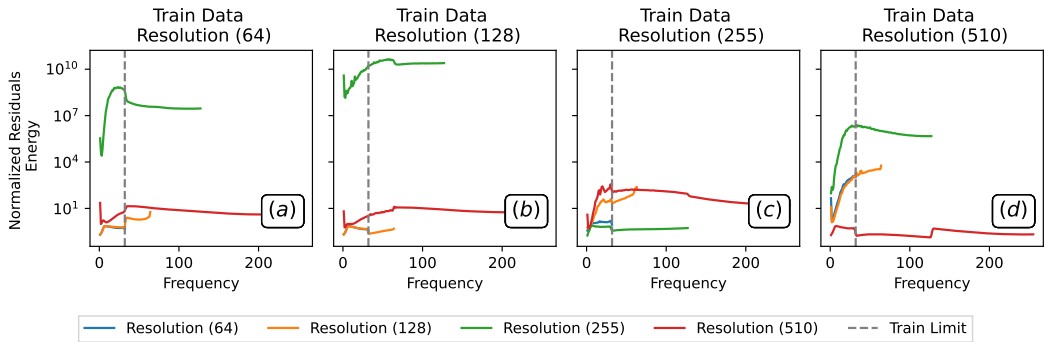

Figure 12: **Interpolation.** Four FNOs trained on Navier Stokes data low-pass limit $32f$ (constant frequency information) and down sampled to resolutions $\{64,128,255,510\}$ (varying sampling rate) from left to right. We test if each model can generalize to data with varying sampling rate. We visualize the normalize residual spectra across test data. Notice that the residual spectra (error) increases substantially in the low frequencies. Lower energy at all wave numbers is better.

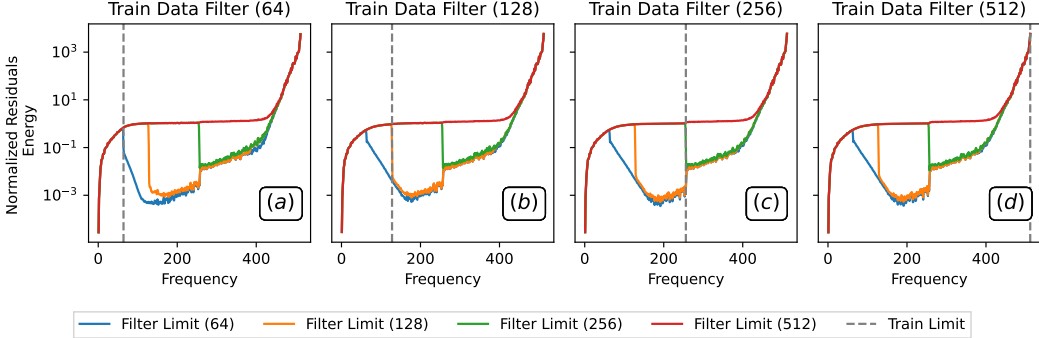

Figure 13: **Extrapolation.** Four FNOs trained on Burgers data of resolution 1024 (constant sampling rate) and low-pass filtered with limits $\{64,128,256,512\}f$ (varying frequency information) from left to right. We test if each model can generalize to data with varying frequency information. We visualize the normalize residual spectra across test data. Notice that the residual spectra (error) increases substantially in the high frequencies. Lower residual energy at all wave numbers is better.

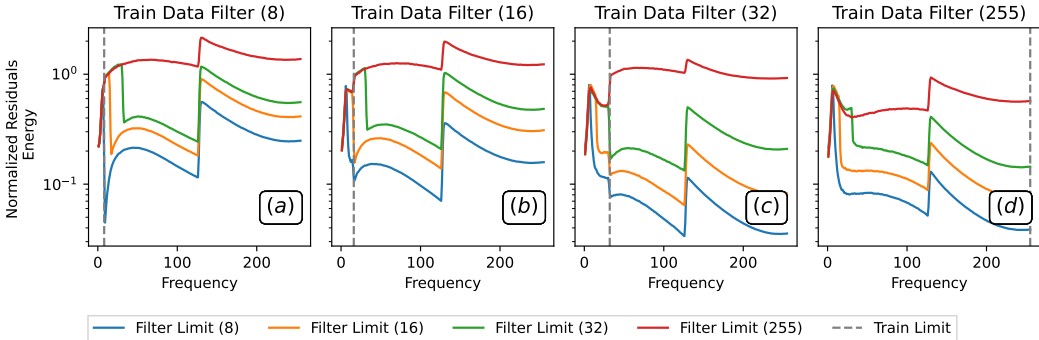

Figure 14: **Extrapolation.** Four FNOs trained on Navier Stokes data of resolution 510 (constant sampling rate) and low-pass filtered with limits $\{8,16,32,255\}f$ (varying frequency information) from left to right. Test if each model can generalize to data with varying frequency information. We visualize the normalize residual spectra across test data. Notice that the residual spectra (error) increases substantially in the high frequencies. Lower residual energy at all wave numbers is better.

# D  EVALUATING SUB- AND SUPER-RESOLUTION

Here we included the full experiment results of studying FNOs' abilities to do zero-shot multi-resolution inference as described in Sec. 3.

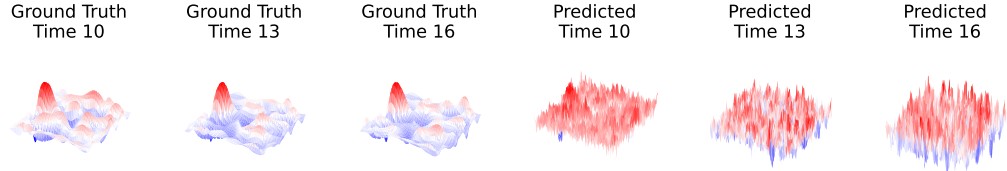

Figure 15: **Aliasing artifacts compound over time.** FNO trained on resolution 255 Navier-Stokes data, evaluated at resolution 510. **Left:** Ground truth evolution of NS fluid flow. **Right:** Corresponding FNO predictions at resolution 510. Notice that the high frequency artifacts become more prevalent over time.

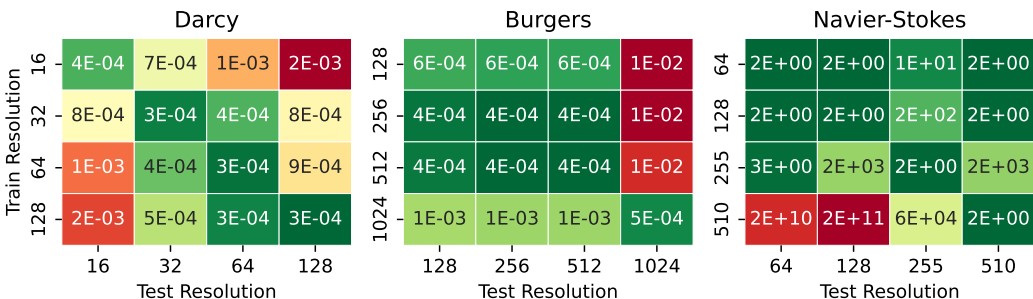

Figure 16: **FNOs do not reliably generalize to higher or lower resolutions.** Heatmaps of losses incurred by FNO trained and tested at varying resolutions (lower is better). When the test resolution varies from the training resolution, the models often incur a substantial increase in loss.

# E    PHYSICS-INFORMED OPTIMIZATION

Here, we use the physics losses of Li et al. (2024b) which explicitly enforce that the governing PDE is satisfied. The governing PDEs are detailed in Appendix A.

Below we include the results of tuning the physics loss weighting coefficient $w$ in Figs. 18 and 17. We then include the full comparisons of training each dataset (Darcy, Burgers, Navier Stokes) at each resolution with both $w \in \{0, 0.1\}$ in Figs. 19-20.

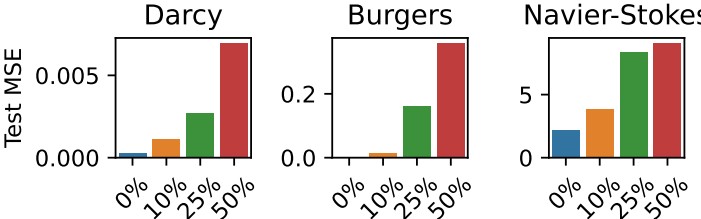

Figure 17: **(Physics-Informed) Optimization.** Increasing the proportion of physics-informed loss in the optimization objective corresponds to increased test loss. Lower MSE is better. Darcy trained at resolution 64, Burgers trained at resolution 512, and Navier Stokes trained at resolution 255.

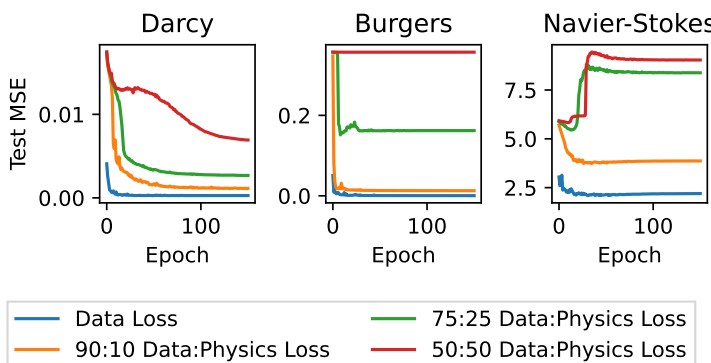

Figure 18: **(Physics-Informed) Optimization Objective.** Physics informed constraints never achieves better performance than using pure data-driven constraints. Darcy trained at resolution 64, Burgers trained at resolution 512, and Navier Stokes trained at resolution 255.

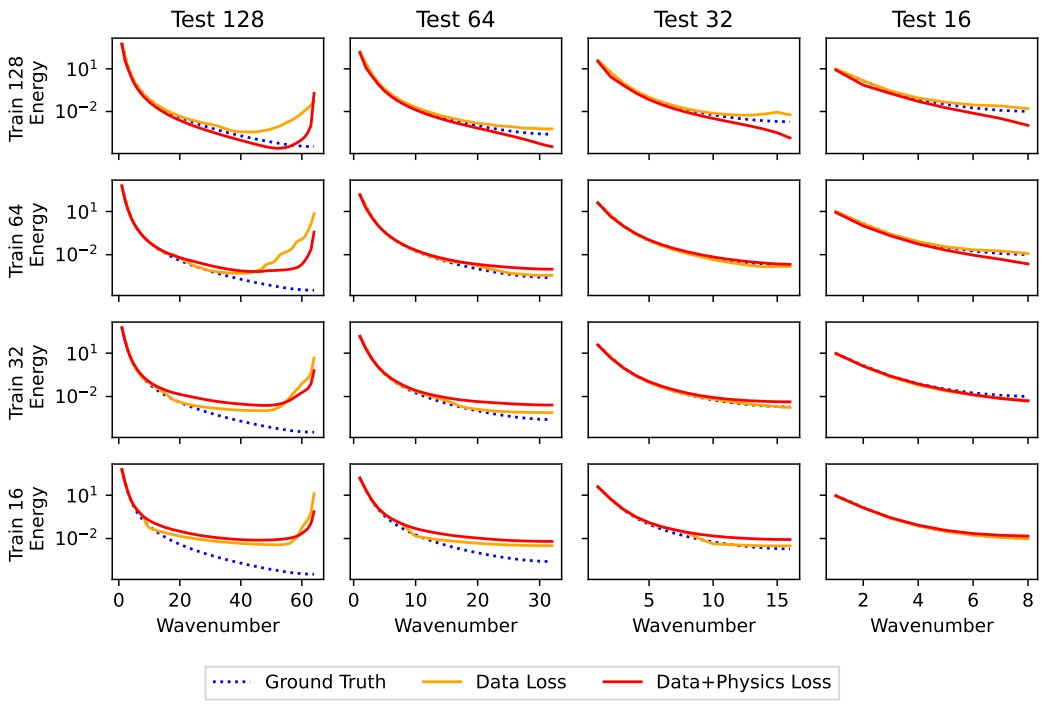

Figure 19: **Darcy.** Energy spectra for models trained at a specific resolution (y-axis) and tested at multiple resolution (x-axis), with and without physics optimization constraint. The spectra generated by the models trained with physics+data loss do not match ground truth. We set $w = 10\%$.

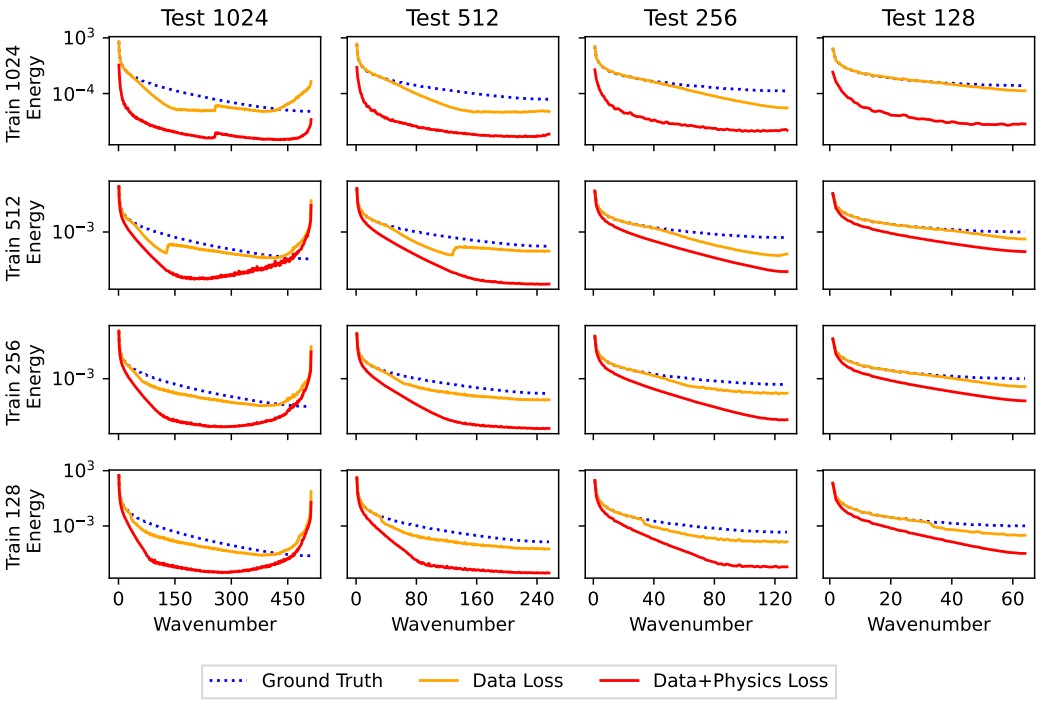

Figure 20: **Burgers.** Energy spectra for models trained at a specific resolution (y-axis) and tested at multiple resolution (x-axis), with and without physics optimization constraint. The spectra generated by the models trained with physics+data loss do not match ground truth. We set $w = 10\%$.

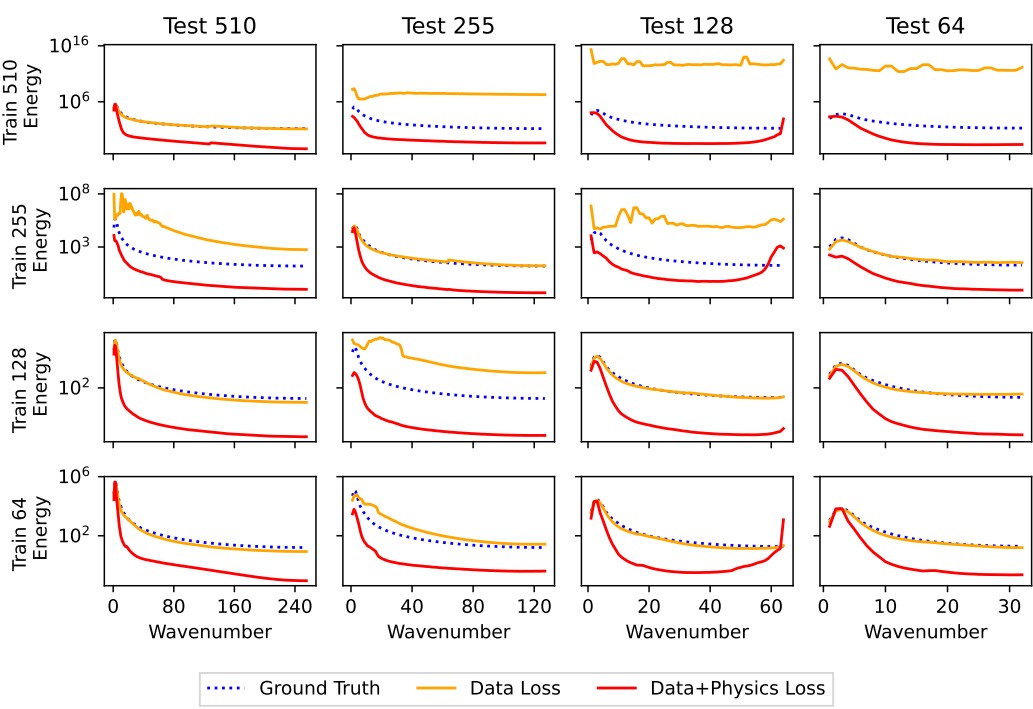

Figure 21: **Navier Stokes.** Energy spectra for models trained at a specific resolution (y-axis) and tested at multiple resolution (x-axis), with and without physics optimization constraint. The spectra generated by the models trained with physics+data loss do not match ground truth. We set $w = 10\%$.

## F    BAND-LIMITED LEARNING

Here, we include the full experimental results of training models with mixed resolution datasets, as described in Sec. 4.2. Note that Gao et al. (2025) did not release a 1D CROP pipeline; therefore, we were unable to test CROP with Burgers.

First, we observe that band-limited learning, in which models are able to both infer and learn over band-limited representations of data, accurately learns the frequency region of the data included in the band-limit. However, these models struggle to learn / do not learn anything outside this range (see Figs. 7, 26, 24). The implication of this is that band-limited-approaches suffice for modeling data within a prespecified range, as long as the band-limit range is wide enough, and the model will never need to be used to infer on data containing additional frequency information. However, we observe the band limited approach **under performs** multi-resolution training at fitting the full spectrum in the multi-resolution inference setting; this is because the resolution of data, and consequently the resolved frequencies, are changing (Fig. 9). For both Darcy (Fig. 24) and Burgers (Fig. 25), we notice that multi-resolution training out-performs band-limited approaches. In Fig. 7, we observe that the predetermined band-limit leads to error in the high frequency range.

A scenario in which CNO and CROP *appear* to perform well is on datasets in which the majority of the energy is concentrated in the predetermined band-limit (e.g., Navier Stokes, see Fig. 10). In this setting, we see that band-limited fit the lower frequencies in the spectrum very well (Fig. 26). However, we also see in Fig. 26, that multi-resolution training is the only method that consistently predicts both the correct amount of energy across the full spectrum. Band limited approaches *fail* to fit the high frequency range of the spectrum.

We note that band-limited approaches do not accurately fit frequencies outside of their predetermined limit. This makes them effective for fixed-resolution inference, but they are *ineffective* for multi-resolution inference. Alternatively, we demonstrated in Sec. 5 that multi-resolution inference can be scaled to new data resolutions (and therefore new parts of the spectral energy spectrum) via scaling up representative samples in the training dataset. We conclude that multi-resolution training is a more flexible and scalable approach to enabling accurate multi-resolution training at *all* parts of the energy spectrum.

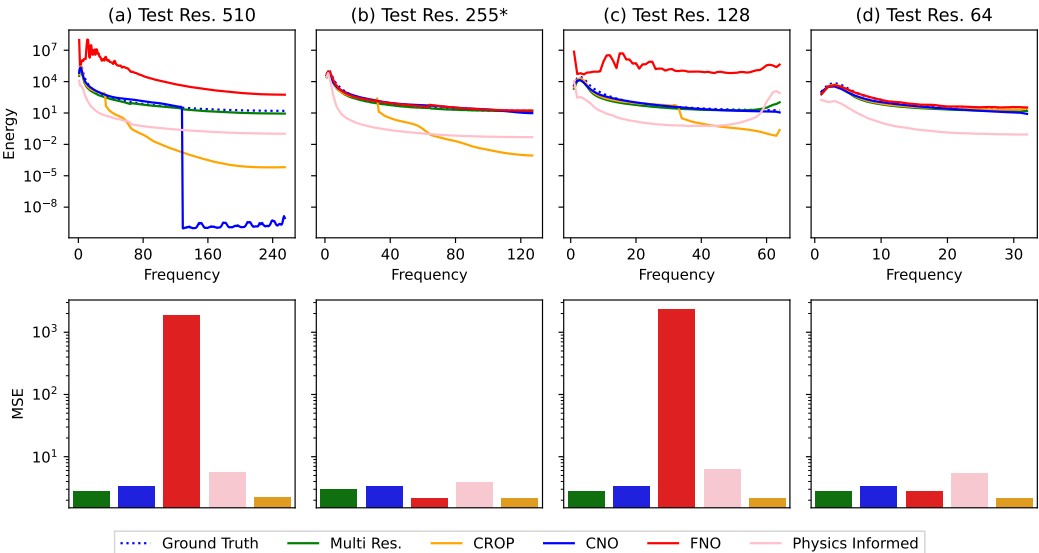

Figure 22: **Spectral Comparison Navier Stokes. Top Row:** Average predicted spectra for test data at varying resolutions across all methods. **Bottom row:** Average mean squared error loss over test data at varying resolutions across all methods. **Zero-shot methods:** CNO, FNO, Physics Informed and CROP are all zero-shot methods, meaning there are trained at a specific resolution (255, indicated by *), and evaluated at resolutions 510, 255, 128, 64. **Data-driven method:** Multi-resolution training; notice that multi-resolution training is the only method that consistently fits both the high and low parts of the spectra. **Band-limited methods:** CNO and CROP are both band-limited methods which are trained in a zero-shot manner at a fixed resolution; we observe that they only accurately fit the low frequencies.

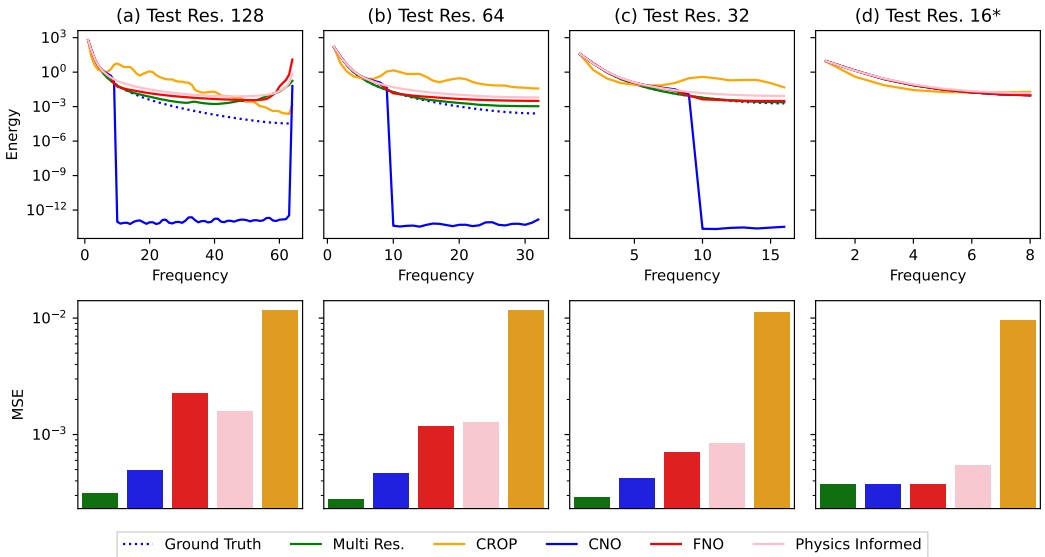

Figure 23: **Spectral Comparison Darcy. Top Row:** Average predicted spectra for test data at varying resolutions across all methods. **Bottom row:** Average mean squared error loss over test data at varying resolutions across all methods. **Zero-shot methods:** CNO, FNO, Physics Informed and CROP are all zero-shot methods, meaning there are trained at a specific resolution (16, indicated by *), and evaluated at resolutions 128, 64, 32, 16. **Data-driven method:** Multi-resolution training; notice that multi-resolution training is the only method that consistently fits both the high and low parts of the spectra. **Band-limited methods:** CNO and CROP are both band-limited methods which are trained in a zero-shot manner at a fixed resolution; we observe that they only accurately fit the low frequencies.

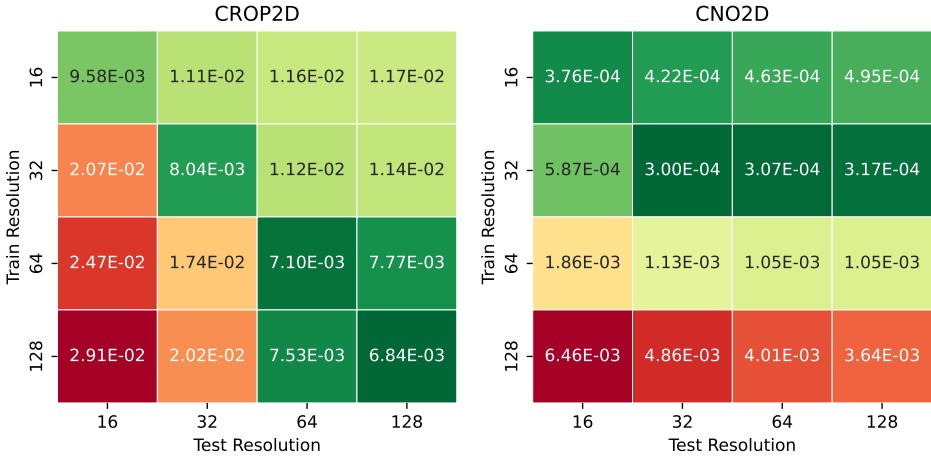

Figure 24: **CROP+FNO and CNO trained on Darcy.** On average, both CROP+FNO and CNO incur higher losses across resolutions compared to both FNO (Fig. 16) and multi-resolution training (Fig. 9).

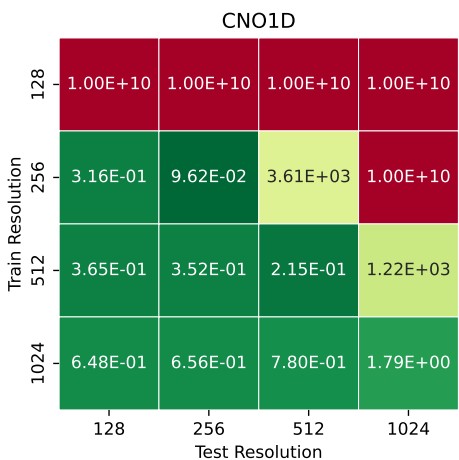

Figure 25: **CNO trained on Burgers.** On average, CNO incur higher losses across resolutions compared to both FNO (Fig. 16) and multi-resolution training (Fig. 9). We note that despite our hyperparameter search (Tab. 2) the CNO model trained on resolution 128 failed to converge.

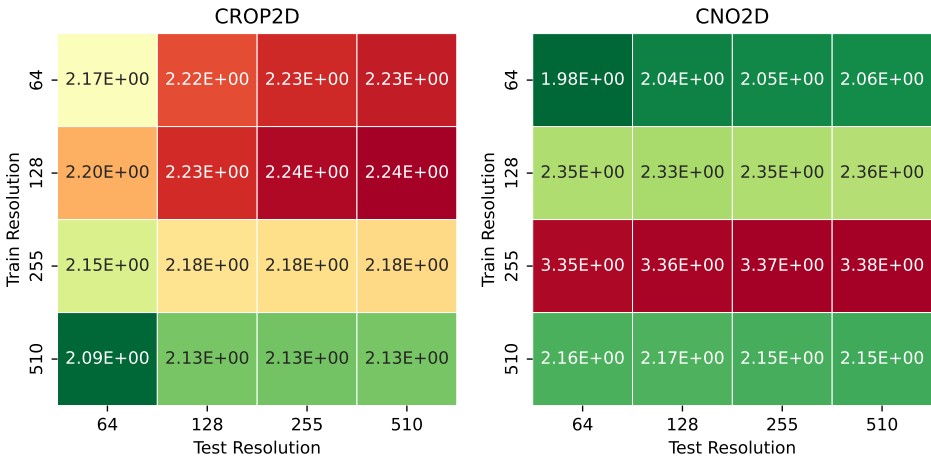

Figure 26: **CROP+FNO and CNO trained on Navier Stokes.** On average, both CROP+FNO and CNO incur lower losses across resolutions compared to both FNO (Fig. 16) and multi-resolution training (Fig. 9).

# G   MULTI-RESOLUTION TRAINING

Here, we include the full experimental results of training models with mixed resolution datasets, as described in Sec. 5. We provide an overview comparison across methods in Fig. 27. See Figs. 28-30 for our main results. We plot the association between increased dataset size and increased training time in Fig. 32.

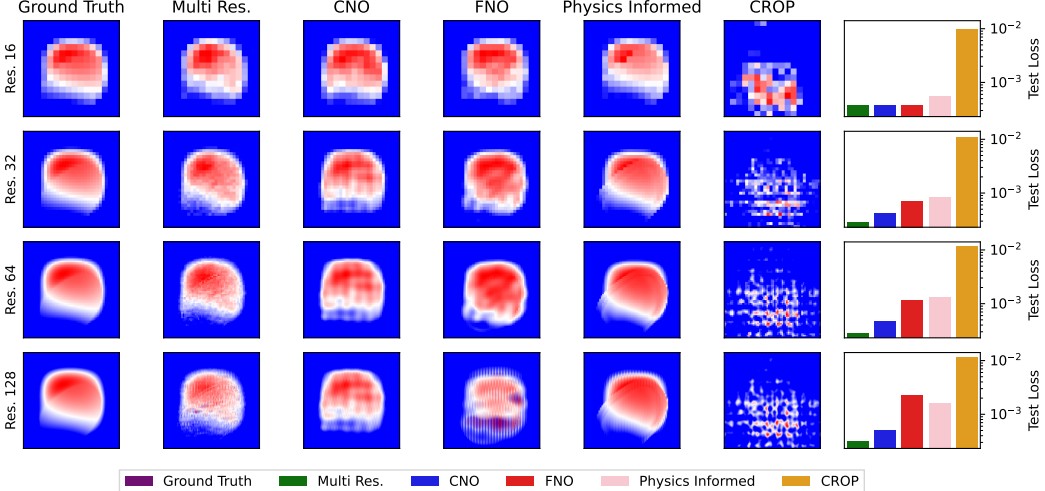

Figure 27: **Assessing multi-resolution inference.** Column 1: Expected prediction for Darcy flow at varying resolutions. Columns 2-6: Sample prediction for Darcy flow at varying test resolutions. Column 7: Average mean squared error test loss at each resolution (lower is better). **Zero-shot methods:** CNO, FNO, Physics Informed and CROP are all zero-shot methods, meaning the model was trained at a specific resolution (16) and evaluated at resolutions 16, 32, 64, 128. **Data-driven method:** Multi-resolution training; notice that multi-resolution training consistently outperforms zero-shot methods.

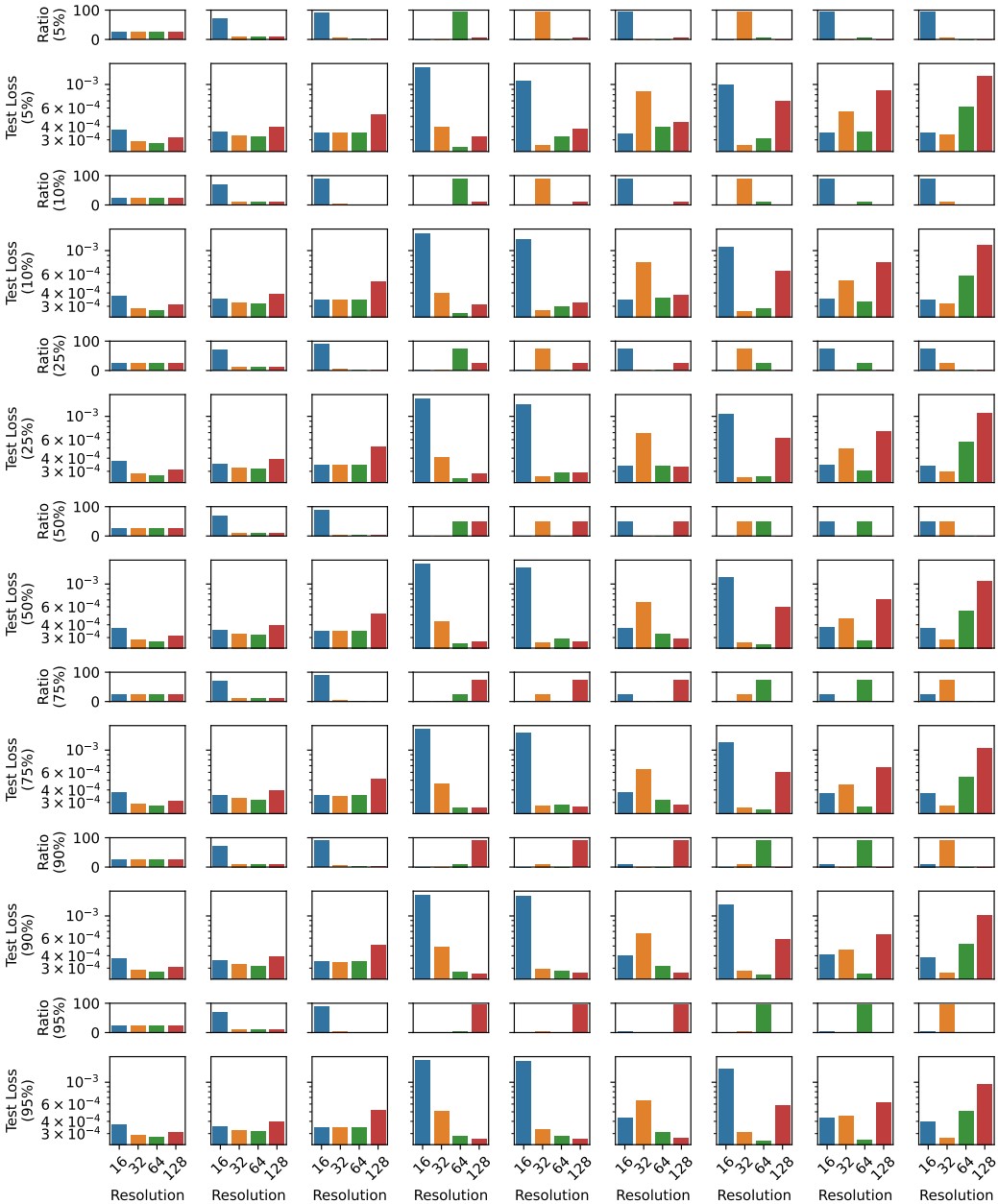

Figure 28: **Darcy Multi-Resolution Training.** FNO trained on multi-resolution Darcy data. Each row include two sub-rows; each row is delineated the the dual-resolution training ratio (indicated in y-axis label). The top sub-row illustrates the ratios of data within each resolution bucket. The bottom sub-row indicates the average test loss across different resolutions. Lower loss is better. Notice in the mixed resolution datasets achieve both low average data size and low loss (ideal scenario).

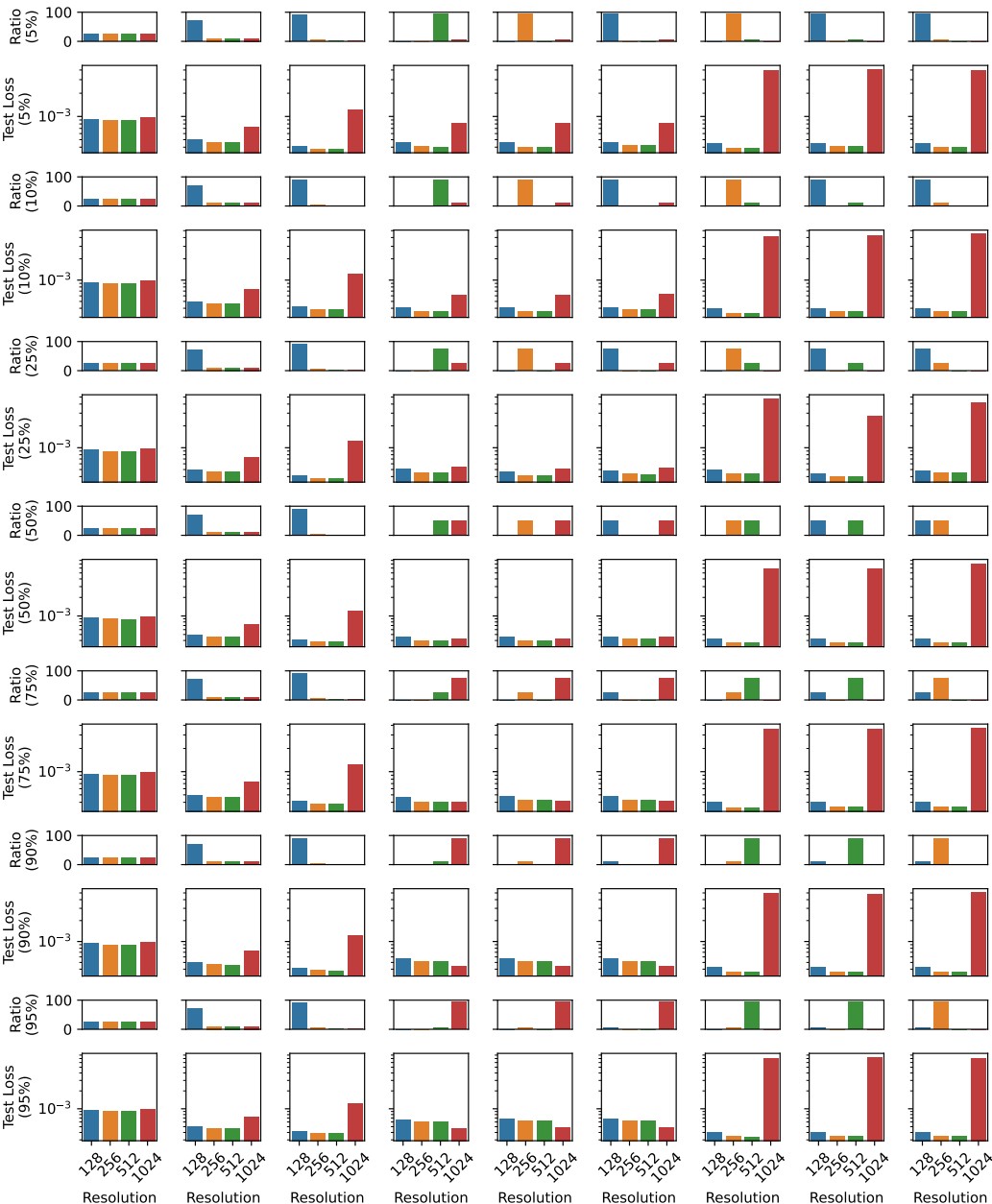

Figure 29: **Burgers Multi-Resolution Training.** FNO trained on multi-resolution Burgers data. Each row include two sub-rows; each row is delineated the the dual-resolution training ratio (indicated in y-axis label). The top sub-row illustrates the ratios of data within each resolution bucket. The bottom sub-row indicates the average test loss across different resolutions. Lower loss is better. Notice in the mixed resolution datasets achieve both low average data size and low loss (ideal scenario).

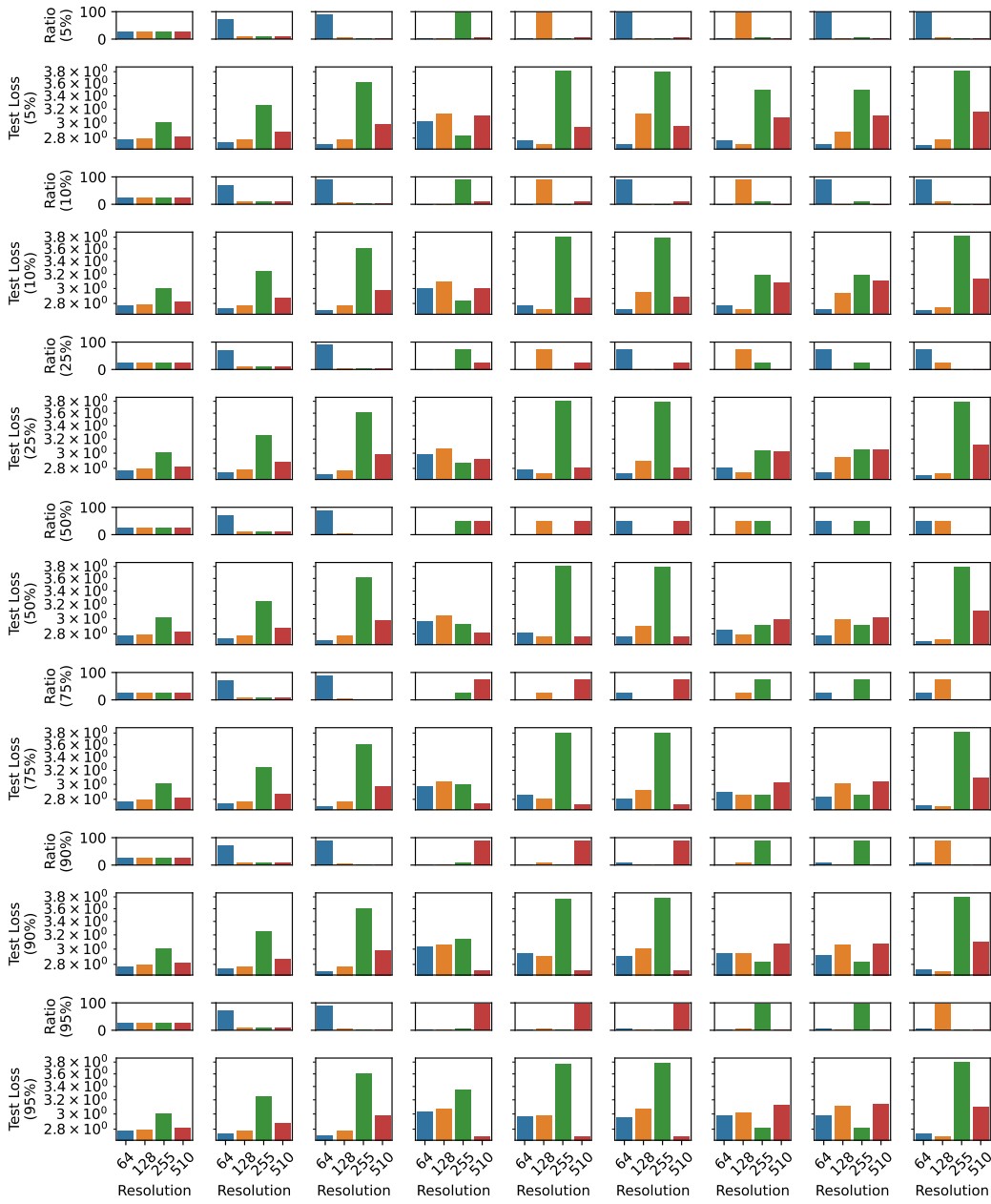

Figure 30: **Navier Stokes Multi-Resolution Training.** FNO trained on multi-resolution Navier Stokes data. Each row include two sub-rows; each row is delineated the the dual-resolution training ratio (indicated in y-axis label). The top sub-row illustrates the ratios of data within each resolution bucket. The bottom sub-row indicates the average test loss across different resolutions. Lower loss is better. Notice in the mixed resolution datasets achieve both low average data size and low loss (ideal scenario).

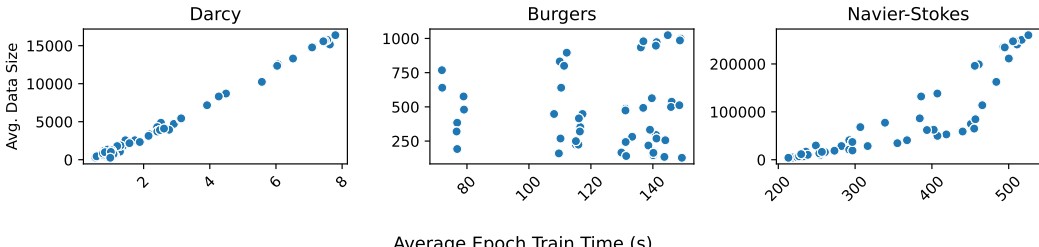

Figure 31: **Data size corresponds with training time**. We notice a clear trend with Darcy and Navier Stokes datasets: as data size increases, so does average training time per epoch. For Burgers, this trend is less clear. However, we note that the Burgers dataset is several orders of magnitude smaller and therefore can be used with a high batch size (see Tab. 1), which reduces the computational gains achieved from a smaller sized dataset.

## G.1    MULTI-RESOLUTION TRAINING PERFORMANCE GAINS

Here we profile both the memory savings (see Tab. 3) and wall-clock training time (see Tab. 4) saving due to the reduction in dataset size from multi-resolution training compared with the single-resolution training counterpart. We empirically observe the smaller dataset size is associated with reduction in training time in Fig. 32.

Table 3: **Impact of Multi-Resolution Training on Memory.** Reporting the size of datasets in Gigabytes (GB) for multi-resolution dataset with ratios (0.9, 0.5, 0.3, 0.2) vs. single-resolution maximum resolution dataset along with the percent decrease in dataset size. The smaller the dataset, the better.

| Dataset | Single-Res. Size (GB) | Multi-Res. Size (GB) | % ($\downarrow$) |
|---|---|---|---|
| Darcy | 0.54 | 0.02 | 96% $\downarrow$ |
| Burgers | 0.34 | 0.005 | 98% $\downarrow$ |
| Navier Stokes | 0.96 | 0.04 | 96% $\downarrow$ |

Table 4: **Impact of Multi-Resolution Training on Wall-clock Training Time.** Reporting the training time (over 150 epochs) for multi-resolution dataset with ratios (0.9, 0.5, 0.3, 0.2) vs. single-resolution maximum resolution dataset along with the percent decrease in dataset size. The lower the training time, the better.

| Dataset | Single-Res. Training Time (hours) | Multi-Res. Training Time (hours) | % ($\downarrow$) |
|---|---|---|---|
| Darcy | 0.33 | 0.05 | 86% $\downarrow$ |
| Burgers | 6.03 | 5.8 | 3.2% $\downarrow$ |
| Navier Stokes | 21.9 | 9.6 | 56% $\downarrow$ |

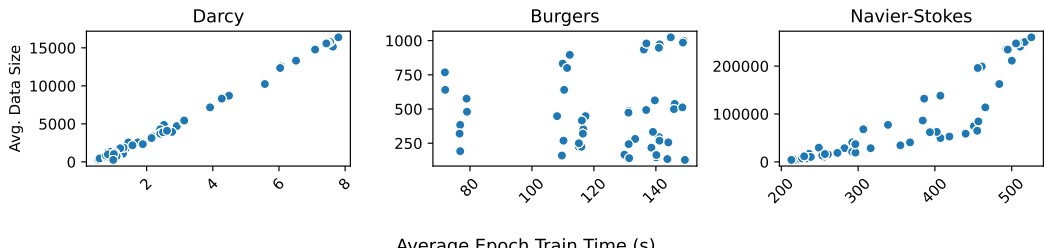

Figure 32: **Data size corresponds with training time**. We notice a clear trend with Darcy and Navier Stokes datasets: as data size increases, so does average training time per epoch. For Burgers, this trend is less clear; however, we note that the Burgers dataset is several orders of magnitude smaller and therefore can be used with a high batch size (see Tab. 1) which reduces the computational gains achieved from a smaller sized dataset.

## H  MODEL IMPLEMENTATIONS AND CONFIGURATIONS

The **Fourier Neural Operator** is described in detail in (Li et al., 2020a); we closely follow their implementation which can be found at `https://neuraloperator.github.io/dev/index.html`. We detail key model configuration parameter choices here:

1. **Max Modes:** When training and evaluating models to do zero-shot super-resolution or sub-resolution and multi-resolution training/inference, we train with the max modes parameter set to half the maximum training resolution. Further, we do an ablation study of the max modes parameter in Appendix I to assess impact of FNO's ability to do information extrapolation and resolution interpolation.
2. **Layer count:** 4 (standard implementation)
3. **Hidden channels:** 32 (standard implementation)
4. **Activation Function:** All main paper experiments utilize the Gelu nonlinearity (Hendrycks, 2016). Further, we assess the impact of utilizing anti-aliasing activation function on zero-shot super and sub-resolution, information extrapolation, and resolution interpolation in Appendix K (Karras et al., 2021).
5. **All other parameters:** All other FNO parameters were initialized to default values detailed in the standard `neuraloperator` Python package (Kossaifi et al., 2024).

The **CNO** is described in detail in (Raonic et al., 2023); we closely follow their implementation which can be found at `https://github.com/camlab-ethz/ConvolutionalNeuralOperator/tree/main`. We detail key model configuration parameter choices here:

1. **Residual block count:** 6 (standard implementation)
2. **Latent size:** 64
3. **Input Resolution:** Varries based on training data resolution.
4. **All other parameters:** All other CNO parameters were initialized to default values detailed in the official CNO implementation: `https://github.com/camlab-ethz/ConvolutionalNeuralOperator/tree/main`.

The **CROP** pipeline is described in detail in (Gao et al., 2025); we closely follow their implementation for CROP+FNO which can be found at `https://github.com/wenhangao21/ICLR25-CROP/tree/main`. We note that they did not include a 1D CROP implementation, so we exclude evaluation of CROP on the 1D Burgers dataset. We detail key model configuration parameter choices here:

1. **Hidden channels:** 32 (standard implementation)
2. **Latent size:** 64
3. **Max Mode:** 32 (half latent size)
4. **All other parameters:** All other CROP parameters were initialized to default values detailed in the official CROP implementation: `https://github.com/wenhangao21/ICLR25-CROP/tree/main`.

The **DeepONet** model is described in detail in (Lu et al., 2021a); we closely follow the implementation of (Gao et al., 2025) which can be found at `https://github.com/wenhangao21/ICLR25-CROP/tree/main`. We detail key model configuration parameter choices here:

1. **Branch layer count:** 4 (standard implementation)
2. **Trunk layer count:** 4 (standard implementation)
3. **Branch layer width:** 128 (standard implementation)
4. **Trunk layer width:** 128 (standard implementation)
5. **All other parameters:** All other DeepONet parameters were initialized to default values detailed in the official implementation of (Gao et al., 2025): `https://github.com/wenhangao21/ICLR25-CROP/tree/main`.

# I  MAX MODES

A key design decision in the FNO architecture is the parameter $m$ that indicates maximum frequencies to keep along each dimension in the Fourier layer during the forward pass. This has implications both during training (which frequencies are learned in the Fourier layers) and at inference (which frequencies are predicted over in the Fourier layers). In Sec. 3, the FNO is always initialized such that it can make use all frequencies in its input; this is especially critical in the multi-resolution setting where data of varying discretization will have varying frequency information. Here, we study the impact of varying $m$. In the zero-shot multi-resolution inference setting, in Figs. 34 and 33, we find that that across all variation in $m$, the models assign high energy in the high-frequencies (e.g., aliasing). More broadly, we comment that in the context of multi-resolution inference, it does not make sense to set $m$ to a value less than the largest populated frequency in a model input, as it ensures that the model cannot make use that frequency information greater than $m$ in the Fourier layers. In the event frequency information above $m$ is not useful to prediction (e.g., noise), we advocate low-pass filtering and down sampling the data to a more compressed representation of data prior to inference to remove unwanted frequencies and ensure faster inference.

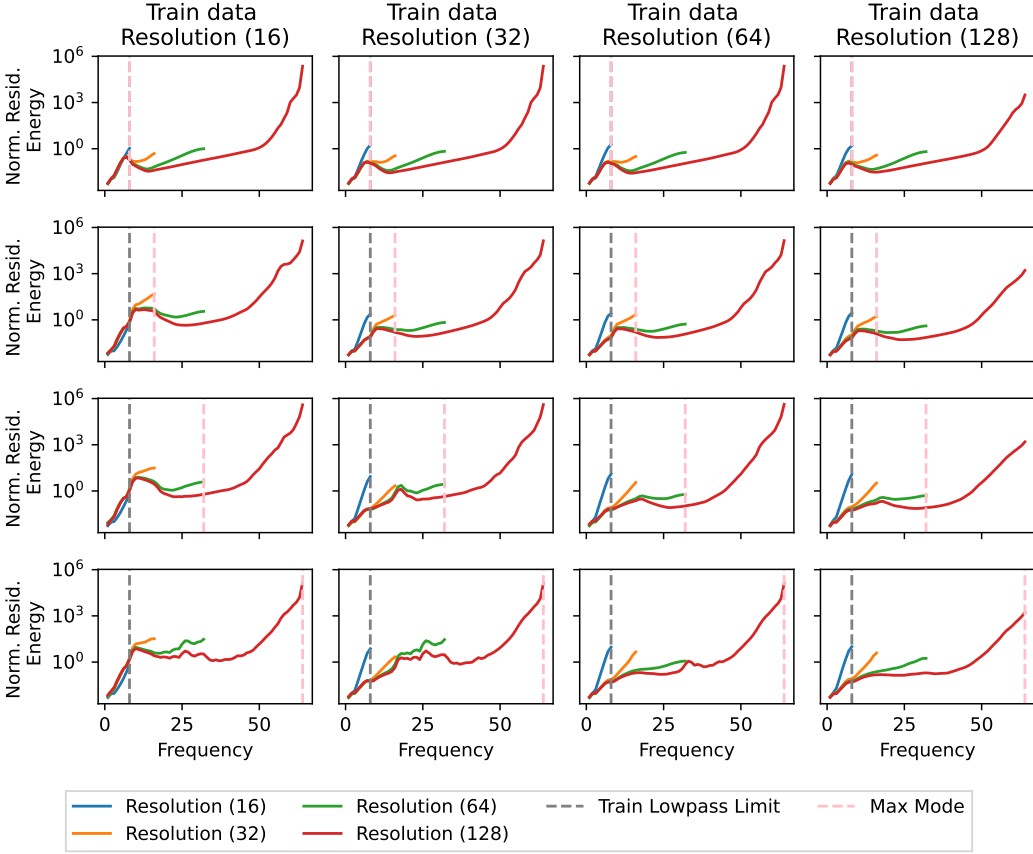

Figure 33: **Resolution Interpolation.** Four FNOs trained on Darcy data low-pass limit = 8 (constant frequency information) and down sampled to resolutions $\{16,32,64,128\}$ (varying sampling rate) from left to right, and top to bottom with max modes $m \in \{8,16,32,64\}$. We test if each model can generalize to data with varying sampling rate. Visualizing the spectrum of the normalized residuals across test data. Notice that the residual spectra (error) increases substantially in the low frequencies. Lower energy at all wave numbers is better. We see that across all variation in $m$, the models assign high energy in the high-frequencies.

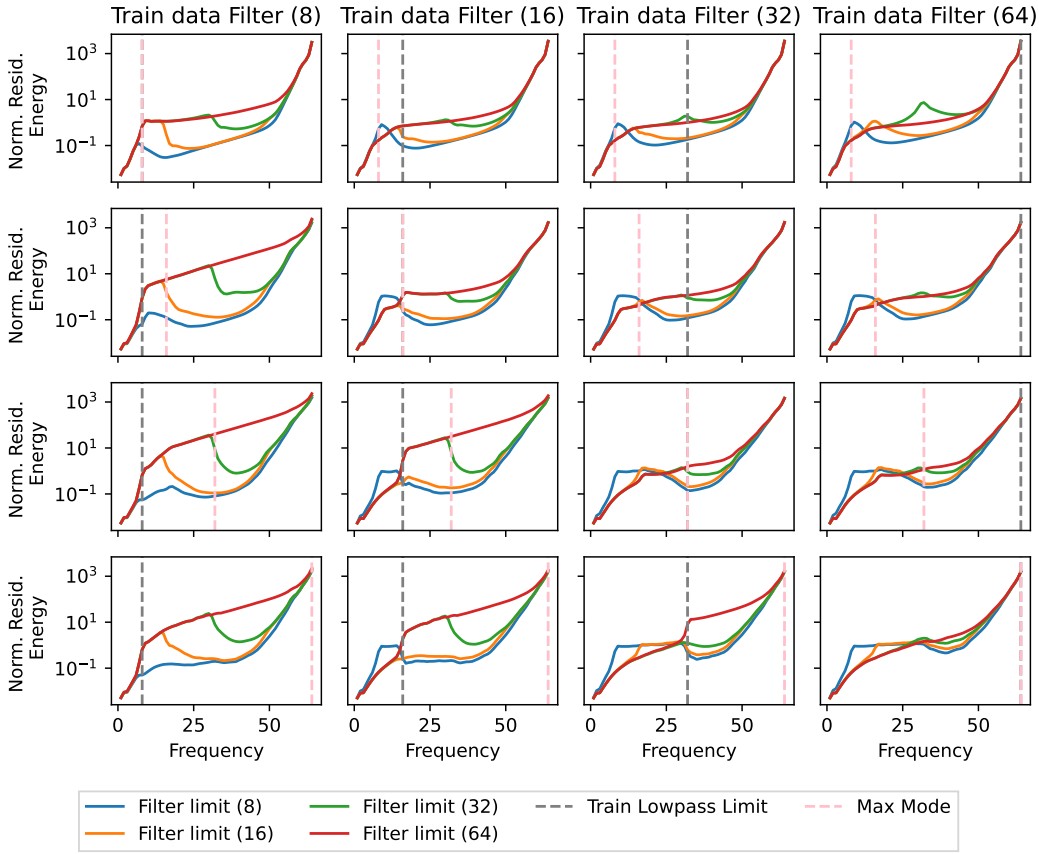

Figure 34: **Information Extrapolation.** Sixteen FNOs trained on Darcy data of resolution 128 (constant sampling rate) and low-pass filtered with limits {8,16,32,64} (varying frequency information) from left to right, and top to bottom with max modes $m \in$ {8,16,32,64}. We test if each model can generalize to data with varying frequency information. Visualizing the spectrum of the normalized residuals across test data. Notice that the residual spectra (error) increases substantially in the high frequencies. Lower residual energy at all wave numbers is better. We observe that across all variation in $m$, the models assign high energy in the high-frequencies.

## J ASSESSING DEEPONET

We extend our investigation to include DeepONet (Lu et al., 2021a). DeepONet is a prominent MLO, however, it notably does not claim to perform accurate **zero-shot** super- and sub-resolution. Despite this, we still investigate the zero-shot super-resolution, sub-resolution, information extrapolation, and resolution interpolation abilities of DeepONet. We then assess if multi-resolution training can enable more accurate multi-resolution inference. Prior to the investigation, we note that DeepONet operates at a fixed resolution on the input and is only resolution independent on the output.

We begin by performing a hyper-parameter search over learning rates $\in \{1e-2, 1e-3, 1e-4, 1e-5\}$; we set a static weight decay $wd = 1e - 5$ as we do not notice significant performance changes w.r.t. weight decay for FNO/CNO/CROP (See Tables 1 and 2). We find the optimal learning rate is $1e - 3$ which we report in Tab. 2.

Next, we assess DeepONet's ability perform information extrapolation and resolution interpolation via experiments detailed in Sec. 3.1 and find that it cannot do either accurately (see Figures 35 and 36). There is little variability in Deponent's information extrapolation performance across test datasets; there is a similar level of aliasing across all test datasets regardless of frequency content.

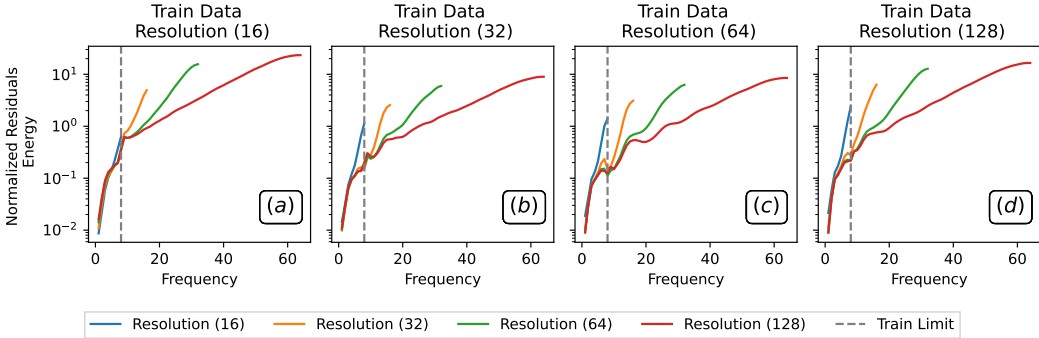

Figure 35: **Resolution Interpolation w/ DeepONet.** Four DeepONets are trained on Darcy data at resolutions $\{16, 32, 64, 128\}$ from left to right with constant frequency information (low-pass limit of $8f$), and are tested on varying resolutions. We assess if each model can generalize to data with varying sampling rate. We visualize spectra of the normalized residuals across test data. Notice, residual spectra (error) increases substantially in the low frequencies. Lower residual energy at all frequencies is better.

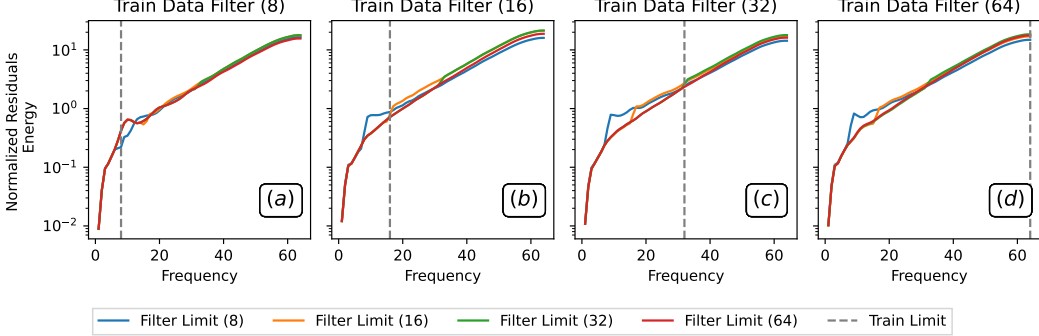

Figure 36: **Information Extrapolation w/ DeepONet.** Four DeepONets trained on Darcy data of resolution 128 (constant sampling rate) and low-pass filtered with limits $\{8, 16, 32, 64\}f$ (varying frequency information) from left to right. Test if each model can generalize to data with varying frequency information. Visualizing spectrum of the normalized residuals across test data. Notice, residual spectra (error) increases substantially in the high frequencies. Lower residual energy at all frequencies is better.

We then assess if DeepONet can perform zero-shot super-resolution and sub-resolution. As we found with other MLOs, we find that it cannot accurately perform zero-shot super-resolution and sub-resolution (See Figures 37 and 38).

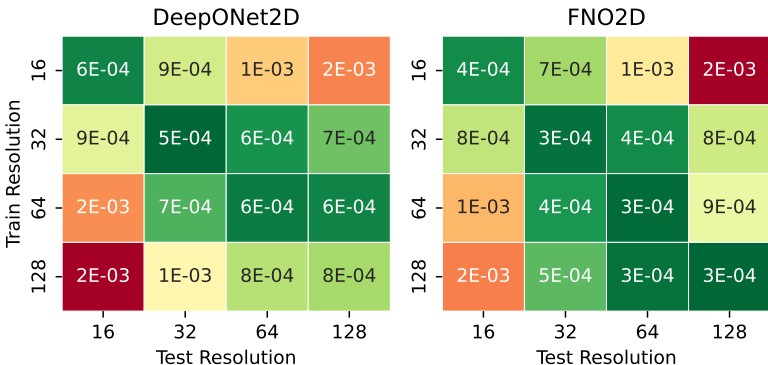

Figure 37: **FNO and DeepONet trained on Darcy.** On average both models incur higher losses across test resolutions that differ from the train resolution. This means both FNO and DeepONet cannot accurately do zero-shot super-resolution or sub-resolution.

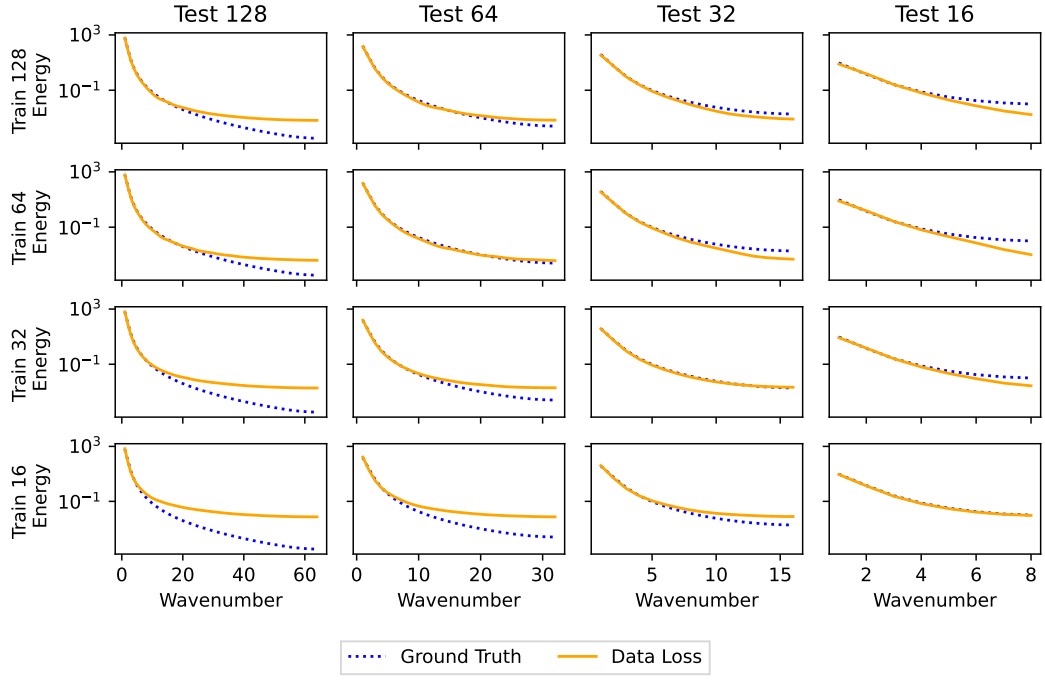

Figure 38: **DeepONet for Darcy.** Energy spectra for models trained at a specific resolution (y-axis) and tested at multiple resolution (x-axis). The spectrums generated by the models do not match ground truth for test resolutions that diverge from the training resolution.

Finally, we assess if multi-resolution training can improve DeepONet's data-driven multi-resolution performance. We employ the same experimental methodology outlined in Sec. 5. We find that multi-resolution training **improves** DeepONet's performance across test resolutions (see Fig. 39). We observe that DeepONet is more sensitive than FNO w.r.t. the ratios of data across resolutions: DeepONet benefits from having more data samples evenly spread out across resolutions. We hypothesize this is due to the fact that DeepONet is only resolution independent on the output, meaning that during training the model must learning mappings from exclusively low resolution inputs to varying resolution outputs (low through high). Because low resolution inputs contain less frequency

components than high-resolution inputs, it is harder learning task than learning between the same resolution input output mappings (as FNO does).

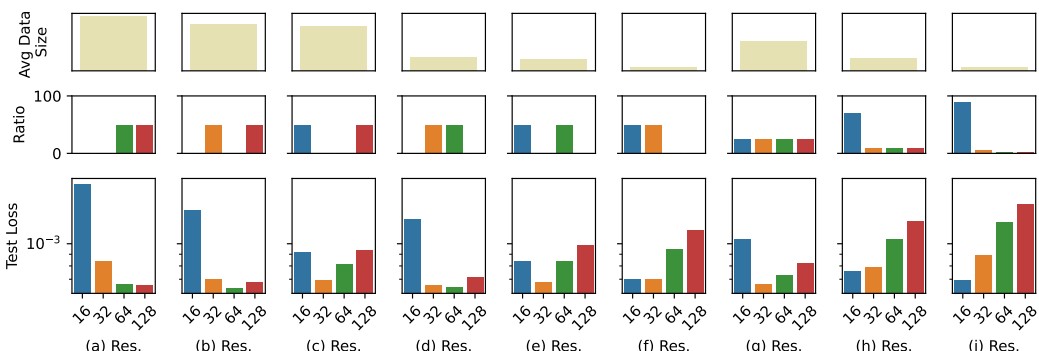

Figure 39: **Multi-resolution training for DeepONet.** DeepONet trained on multi-resolution Darcy data. **Top row:** Average number of pixels in a data sample in the training set. Lower number of pixels enables faster data generation and model training. **Middle row:** The ratios of data within each resolution bucket. **Bottom row:** The average test loss across different resolutions. Lower loss is better. Notice: mixed resolution datasets achieve both low average data size and low loss (ideal scenario).

## K ASSESSING ANTI-ALIAS NONLINEARITIES

We extend our investigation of FNO to assess the impact nonlinearities play on model aliasing in the multi-resolution inference setting. As we discuss in Sec. 6, Karras et al. (2021) observed that the application of point-wise nonlinearities to intermediate model representations introduces high-frequency components that cannot be resolved via the given resolution. To mitigate this source of aliasing, they propose *anti-aliasing* nonlinearities which simply interpolate data to a greater resolution by a scalar factor, apply the desired non-nonlinearity to the higher resolution representation, and then interpolate the signal back down to the original resolution (thus removing many of the introduced high-frequency components). The CNO architecture utilizes these anti-alias nonlinearities. We now investigate the impact of anti-alias nonlinearities to disentangle architecture- vs. data-pipeline-driven aliasing for FNO.

We begin by assessing the FNO with anti-aliasing nonlinearities' ability to perform information extrapolation and resolution interpolation via experiments detailed in Sec. 3.1 and find that it cannot do either accurately (see Figures 40 and 41). This confirms that the out-of-distribution nature of data plays a significant role in aliasing for FNO.

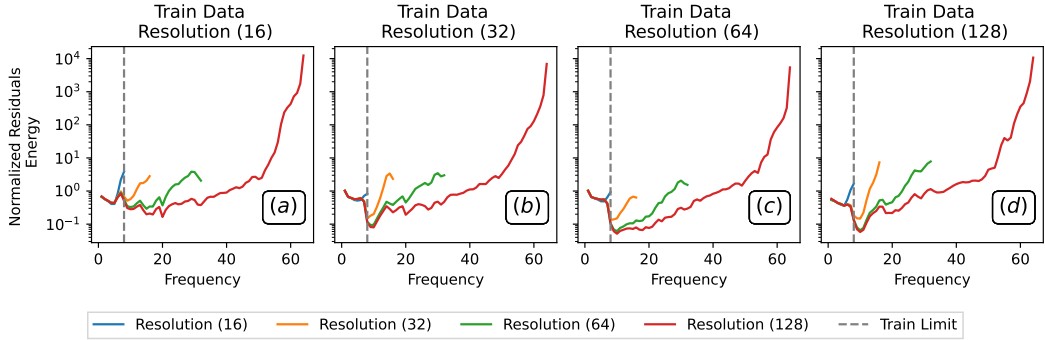

Figure 40: **Resolution Interpolation w/ FNO+Anti-Aliasing Activation.** Four FNOs w/ anti-aliasing activation functions are trained on Darcy data at resolutions $\{16, 32, 64, 128\}$ from left to right with constant frequency information (low-pass limit of $8f$), and are tested on varying resolutions. We assess if each model can generalize to data with varying sampling rate. We visualize spectra of the normalized residuals across test data. Notice, residual spectra (error) increases substantially in the low frequencies. Lower residual energy at all frequencies is better.

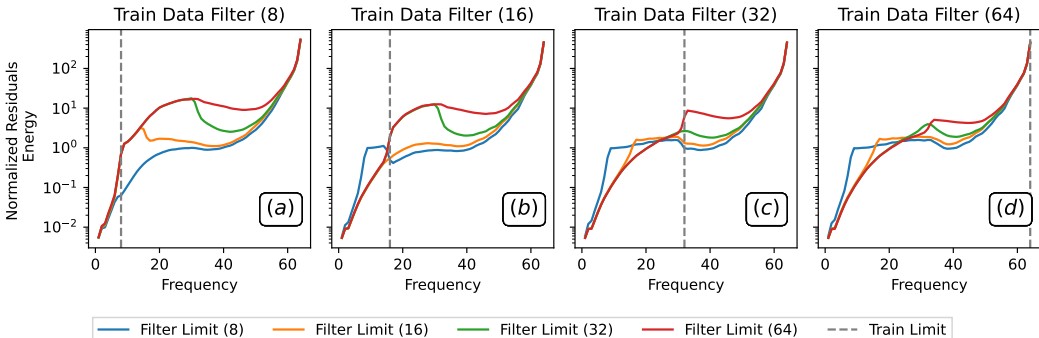

Figure 41: **Information Extrapolation w/ FNO+Anti-Aliasing Activation.** Four FNOs w/ anti-aliasing activation functions are trained on Darcy data of resolution 128 (constant sampling rate) and low-pass filtered with limits $\{8, 16, 32, 64\}f$ (varying frequency information) from left to right. Test if each model can generalize to data with varying frequency information. Visualizing spectrum of the normalized residuals across test data. Notice, residual spectra (error) increases substantially in the high frequencies. Lower residual energy at all frequencies is better.

We then assess if FNO with anti-aliasing nonlinearities can perform zero-shot super-resolution and sub-resolution. As we found with other MLOs, we find that it cannot accurately perform zero-shot super-resolution and sub-resolution (See Figures 42 and 43). Therefore, we conclude that while anti-aliasing nonlinearities stem architecture-driven aliasing artifacts in the model forward pass, they do not address the data-driven aliasing artifacts introduced during inference on data of resolutions that differ from a model's training resolution.

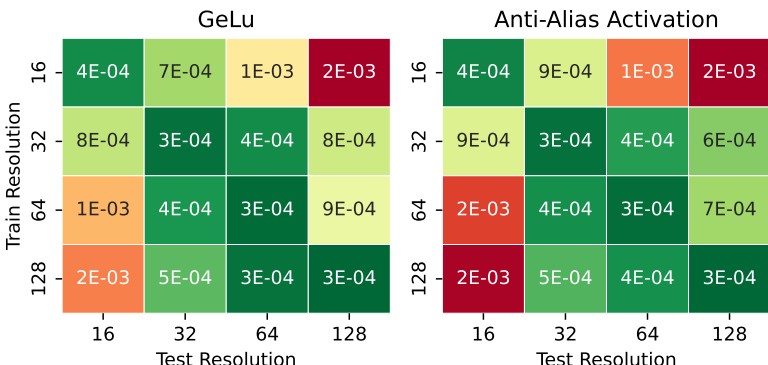

Figure 42: **FNO w/ Anti-Aliasing vs. GeLu activation functions trained on Darcy.** Both models incur higher losses across test resolutions that differ from the train resolution. This means that activation functions alone are not the source of FNO aliasing in the zero-shot multi-resolution setting; this confirms that data at resolutions different than the model's training resolution is sufficiently out-of-distribution. Zero-shot multi-resolution inference is unreliable regardless of activation function choice.

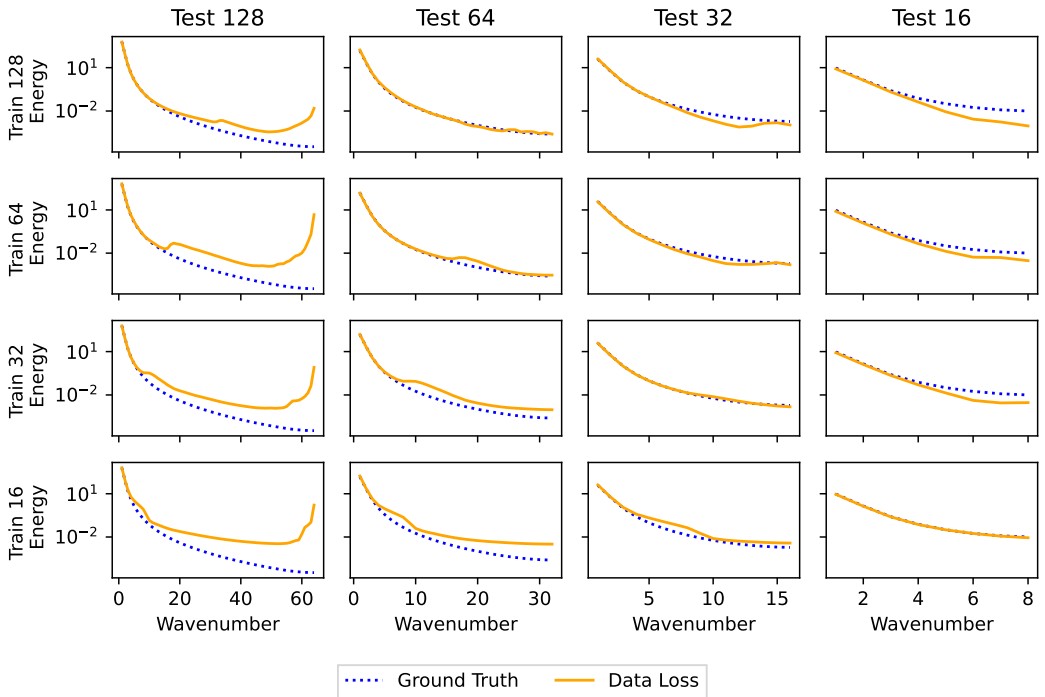

Figure 43: **FNO w/ Anti-Aliasing activation function for Darcy.** Energy spectra for models trained at a specific resolution (y-axis) and tested at multiple resolution (x-axis). The spectrums generated by the models do not match ground truth for test resolutions that diverge from the training resolution.

## L    SENSITIVITY ANALYSIS

We include an extended sensitivity analysis of our resolution interpolation, information extraction and zero-shot super-/sub-resolution experiments for the Darcy dataset. We report the mean and standard deviation over these experiments over three random seeds and show consistent performance across all seeds. In the main paper, we ensured the robustness of our results by replicating observed trends across a wide array of datasets (Darcy, Burgers, Navier-Stokes) as a proxy for generalizability of results. Doing multi-seeded experiments was infeasible for the larger datasets (Burgers and Navier-Stokes) due to computational cost. By including both multi-seeded experiments and experiments over multiple datasets, we ensure that results are generalizable.

We begin by assessing the FNO's ability to perform information extrapolation and resolution interpolation via experiments detailed in Sec. 3.1 over three random seeds and find that it cannot do either accurately (see Figures 44 and 45). Results are consistent over multiple seeds.

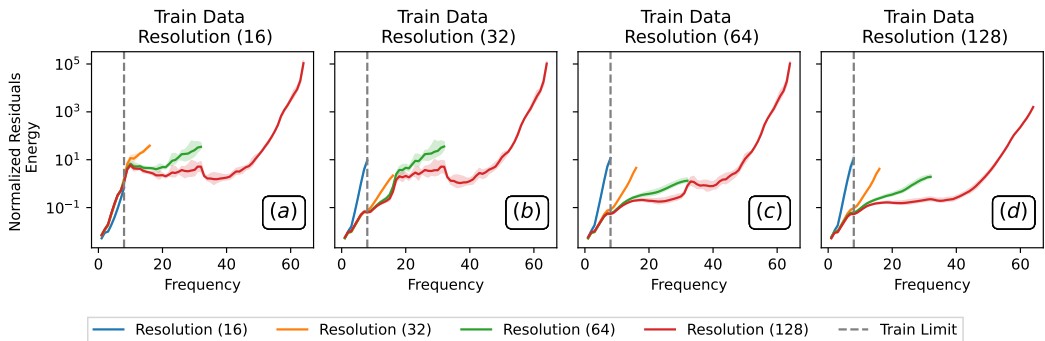

Figure 44: **Resolution Interpolation over 3 Random Seeds.** Four FNOs are trained on Darcy data at resolutions $\{16, 32, 64, 128\}$ from left to right with constant frequency information (low-pass limit of $8f$), and are tested on varying resolutions. We assess if each model can generalize to data with varying sampling rate. We visualize spectra of the normalized residuals across test data. Notice, residual spectra (error) increases substantially in the low frequencies. Lower residual energy at all frequencies is better. Results averaged over three random seeds; shaded region depicts standard deviation.

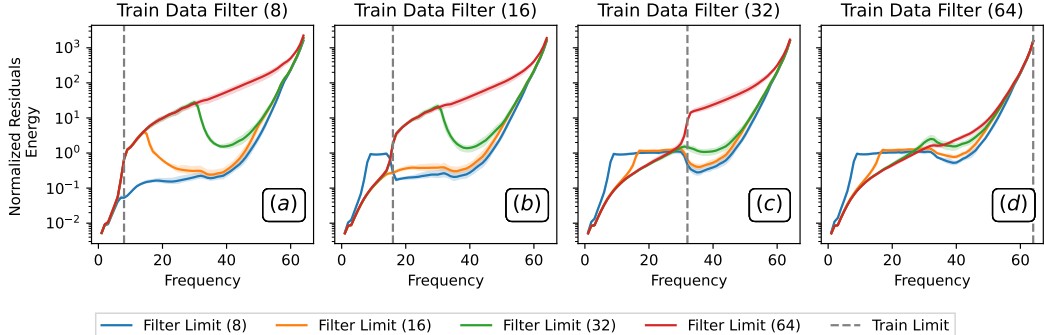

Figure 45: **Information Extrapolation over 3 Random Seeds.** Four FNOs are trained on Darcy data of resolution 128 (constant sampling rate) and low-pass filtered with limits $\{8, 16, 32, 64\}f$ (varying frequency information) from left to right. Test if each model can generalize to data with varying frequency information. Visualizing spectrum of the normalized residuals across test data. Notice, residual spectra (error) increases substantially in the high frequencies. Lower residual energy at all frequencies is better. Results averaged over three random seeds; shaded region depicts standard deviation.

We then assess if FNO can perform zero-shot super-resolution and sub-resolution over three random seeds (see Fig. 46). We observe very similar performance across all seeds: FNO consistently fails

to perform accurate zero-shot super-/sub-resolution. We report mean and standard deviation of this experiment across all three seeds in Tab. 5.

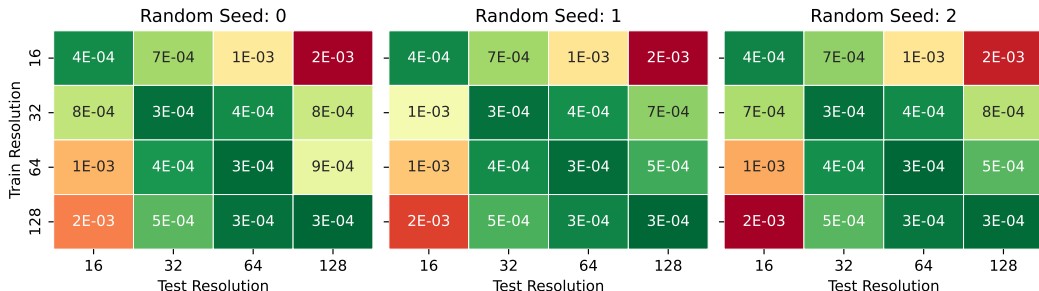

Figure 46: **FNO on Darcy over three random seeds.** On average FNO incurs higher losses across test resolutions that differ from the train resolution across all three random seeds.

Table 5: **Assessing FNO zero-shot super-/sub-resolution over multiple seeds.** Reporting mean and standard deviation of MSE over 3 random seeds.

| Train Res. / Test Res. | 16 | 32 | 64 | 128 |
|---|---|---|---|---|
| 16 | 3.6e-4 $\pm$ 9e-6 | 6.8e-4 $\pm$ 2e-5 | 1.2e-3 $\pm$ 3e-5 | 2.0e-3 $\pm$ 3e-4 |
| 32 | 8.2e-4 $\pm$ 1e-4 | 2.9e-4 $\pm$ 2e-6 | 3.8e-4 $\pm$ 9e-6 | 7.4e-4 $\pm$ 8e-5 |
| 64 | 1.4e-3 $\pm$ 8e-5 | 3.9e-4 $\pm$ 2e-5 | 2.7e-4 $\pm$ 1e-5 | 6.4e-4 $\pm$2e-4 |
| 128 | 1.8e-3 $\pm$ 8e-5 | 5.2e-4 $\pm$ 1e-5 | 3.2e-4 $\pm$ 3e-5 | 2.7e-4 $\pm$ 8e-6 |

## M    ASSESSING DATA DOWN-SAMPLING METHOD

Unless otherwise states we follow the standard PDEBench convention for changing the resolution of our data in our experiments: strided downsampling Takamoto et al. (2022). Put simply, strided downsampling reduced the resolution of a signal by sampling every $Nth$ point; therefore, to reduce the resolution of a signal by 50%, every other point in the signal would be kept (and the rest discarded). We choose strided downsampling as it simulates realistic experimental settings in which you have measurements of a system at fixed intervals and no method to resolve intermediate points between intervals (e.g., sensors for monitoring a real-world system).

We now investigate the impact of sampling strategy by assessing the zero-shot super-/sub-resolution capability of models trained on data downsampled via data striding vs. average pooling vs. bilinear interpolation. In all cases, we consistently observe a failure to accurately fit the high-frequency regime when conducting zero-shot super-resolution (see Figures47-50).

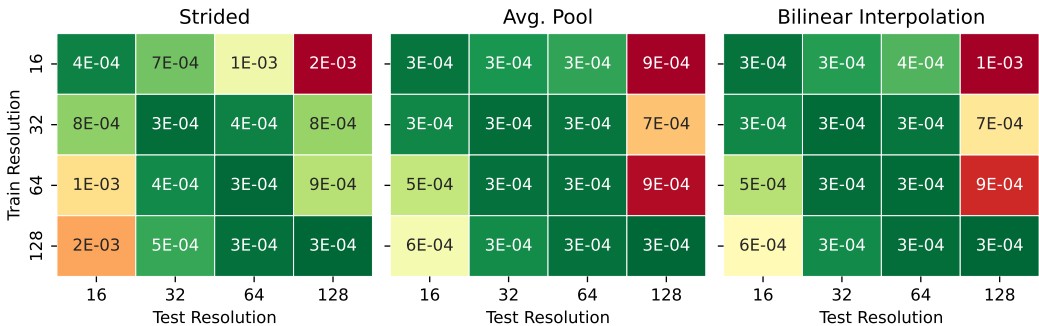

Figure 47: **FNO on Darcy with varied data downsampling strategies.** FNO is trained and tested on data that is downsampled via from the max resolution of 128 via three strategies: strided downsampling, average pooling, and bilinear interpolation. On average FNO incurs higher losses across test resolutions that differ from the train resolution across all three downsampling strategies.

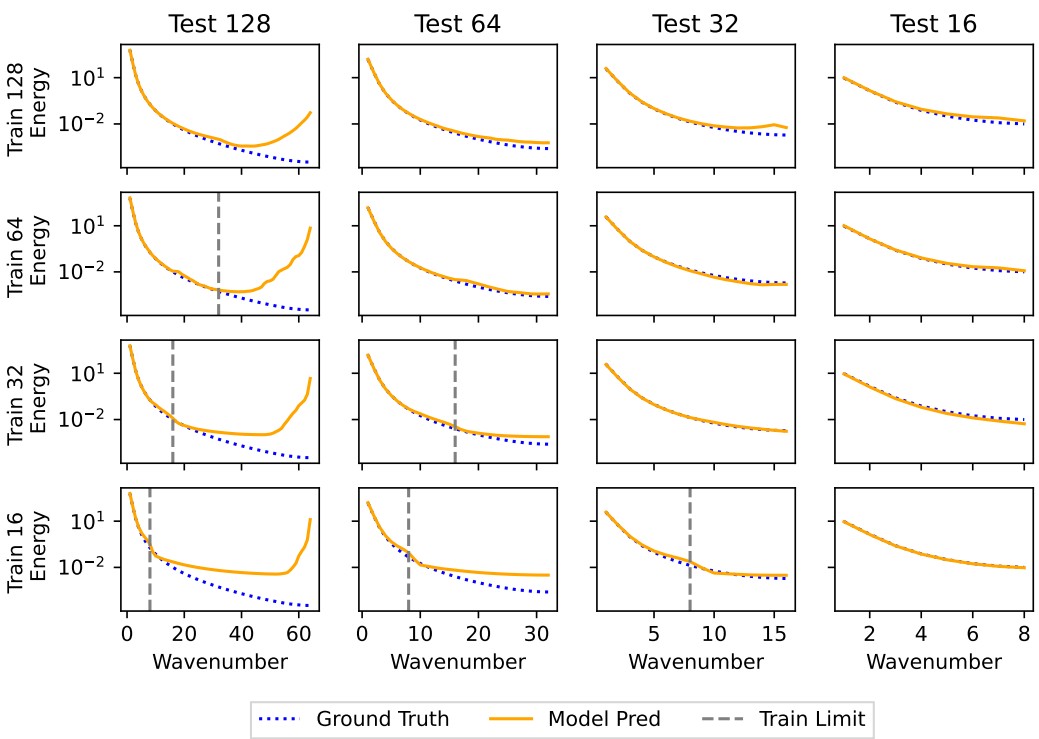

Figure 48: **FNO trained on Darcy (strided downsampling).** Energy spectra for models trained at a specific resolution (y-axis) and tested at multiple resolution (x-axis). The spectrums generated by the models do not match ground truth for test resolutions that diverge from the training resolution. The train and test data that are less than 128 are downsampled via strided downsampling. The gray line indicates the maximum frequency present in the model's training data.

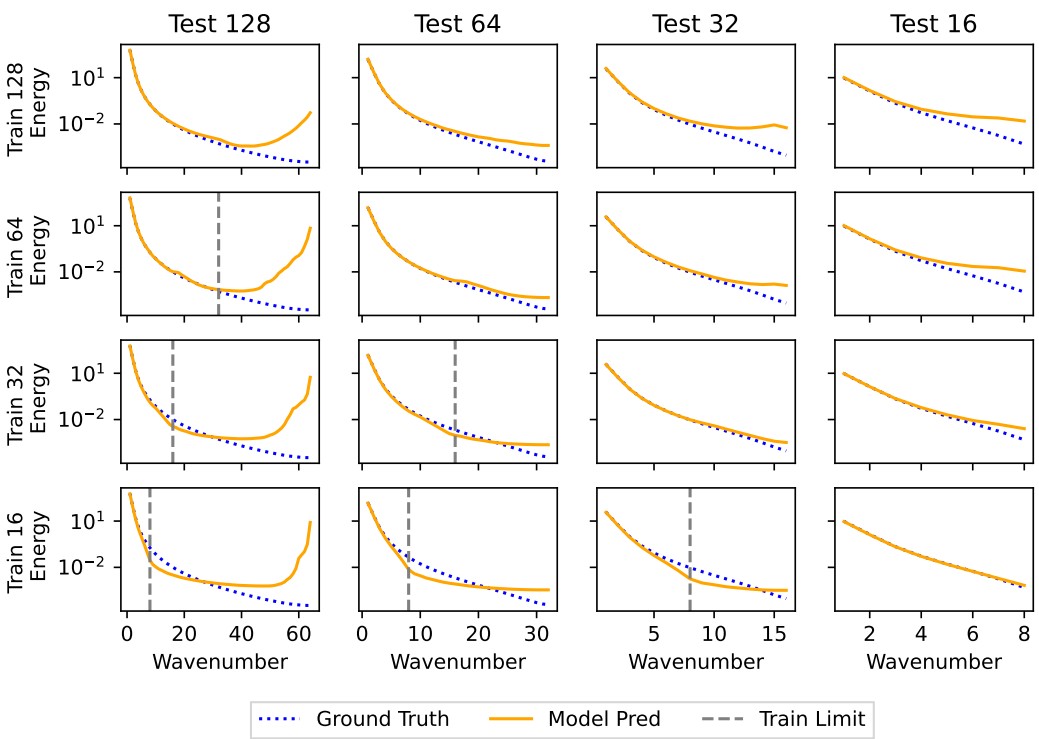

Figure 49: **FNO trained on Darcy (average pooling).** Energy spectra for models trained at a specific resolution (y-axis) and tested at multiple resolution (x-axis). The spectrums generated by the models do not match ground truth for test resolutions that diverge from the training resolution. The train and test data that are less than 128 are downsampled via average pooling. The gray line indicates the maximum frequency present in the model's training data.

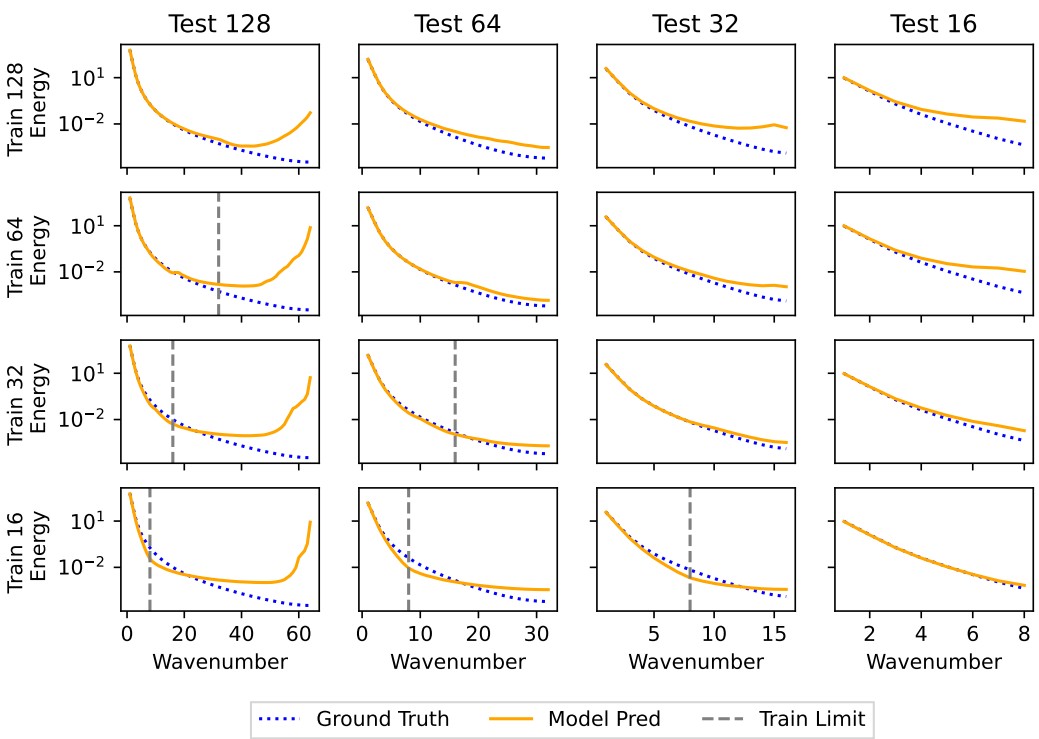

Figure 50: **FNO trained on Darcy (bilinear interpolation).** Energy spectra for models trained at a specific resolution (y-axis) and tested at multiple resolution (x-axis). The spectrums generated by the models do not match ground truth for test resolutions that diverge from the training resolution. The train and test data that are less than 128 are downsampled via bilinear interpolation. The gray line indicates the maximum frequency present in the model's training data.

## N  BENCHMARKING PERFORMANCE ACROSS COMPUTING ARCHITECTURES

We profile the performance of FNO on the super-resolution inference task in both the zero-shot and multi-resolution training setting across multiple computing architectures. In the zero-shot super-resolution task, we train an FNO on resolution 16 Darcy data. In the multi-resolution training-based super-resolution task, we train an FNO on a multi-resolution Darcy dataset comprised of 25% 16 resolution, 25% 32 resolution, 25% 64 resolution, 25% 128 resolution data. We evaluate both models on test data of varying resolutions: 16, 32, 64, 128.

In all experiments, we use Python 3.10 and "neuralopertor" library (Kossaifi et al., 2025) version 2.0.0. Notably, we were not able to acquire access to AMD GPU hardware and older NVIDIA hardware (e.g., GPUs in the NVIDIA RTX/P100 lines which were used in the previous work (Li et al., 2020a) were not available to us). Further, while we did have access to the Intel Data Center GPU Max Series (128GB HBM), the hardware-compatible PyTorch version 2.8.0a0+gitba56102 did not support the full stack of operations used by FNO layers so were were unable to profile performance on that hardware. We tested the follow computing architectures:

1. AMD EPYC 7763 (Milan) CPU; PyTorch version 2.9.1
2. NVIDIA A100-PCIE-40GB with driver version 550.163.01; CUDA Toolkit version 12.4.131; PyTorch version 2.9.1+cu126
3. NVIDIA A40 with driver verion 590.44.01; CUDA Toolkit version 13.1.80; PyTorch version 2.10.0+cu130
4. NVIDIA H100 NVL with driver verion 590.44.01; CUDA Toolkit version 13.1.80; PyTorch version 2.10.0+cu130
5. NVIDIA H200 NVL with driver verion 590.44.01; CUDA Toolkit version 13.1.80; PyTorch version 2.10.0+cu130
6. Apple M2 Pro; Metal 4; PyTorch version 2.10.0

We observe that there is wide variance in error across different hardware (see y-axes on Fig. 51). Despite this, we confirm the core trend of multi-resolution training outperforming zero-shot super resolution is consistent across computing architectures. In Figures 53-57, we note that the predicted spectra of the zero-shot model consistently diverged from the label spectra after frequency 8 (the maximum observable frequency in the training data) across all hardware; this was not the case for the model trained on multi-resolution data. This suggests that variance in hardware is not the reason for poor zero-shot super-resolution, but rather it is the out-of-distribution nature of the zero-shot inference task. While implementation differences in hardware-software stacks give rise to different levels of error in the model's forward pass it does so consistently across different types of model training regimes (e.g., single-resolution vs. multi-resolution). Therefore, we conclude that these hardware performance differences are not key drivers of data-driven aliasing artifacts in FNO.

Finally, we caution users to be aware of the wide-range of FNO performance across different computing architectures and invite the community to further replicate our experiments on additional architectures. To enable this, in addition to releasing our full set of experimental code[1], we release both these hardware tests[2] and a more comprehensive set of tests to reproduce a representative subset of the experiments from this paper[3].

---

[1] https://github.com/msakarvadia/operator_aliasing
[2] https://github.com/msakarvadia/operator_aliasing/blob/main/experiments/multi_architecture_test.py
[3] https://github.com/msakarvadia/operator_aliasing/blob/main/notebooks/demo.ipynb

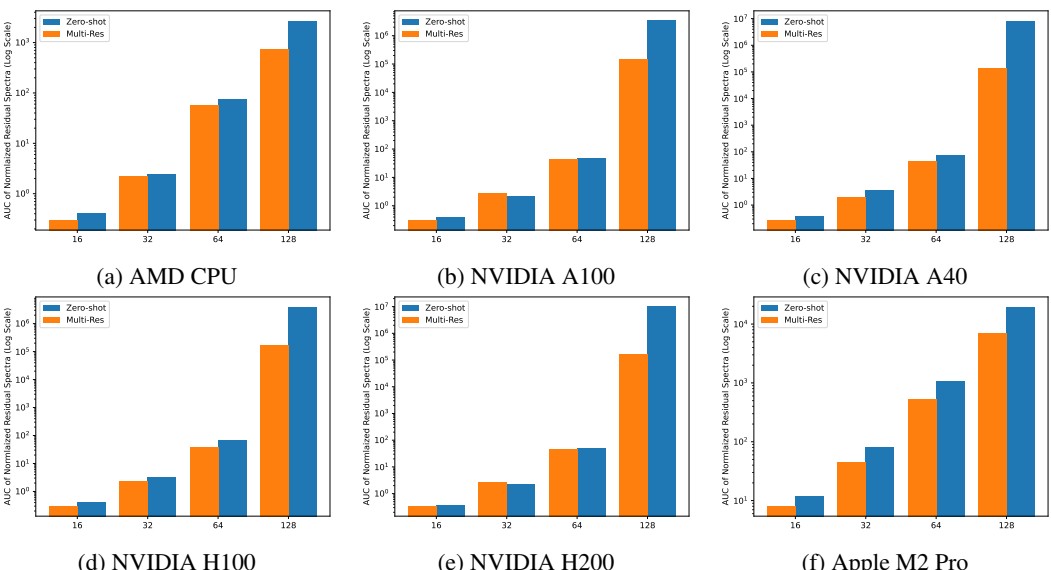

Figure 51: **Zero-Shot v.s. Multi-Resolution Training for Super-Resolution.** X-axis: area under the curve (AUC) of the normalized spectrum of the residuals of the model predictions (lower is better). Y-axis: Test resolution. Across all computing architectures, we notice that the multi-resolution model consistently out-performs the zero-shot model across all test resolutions.

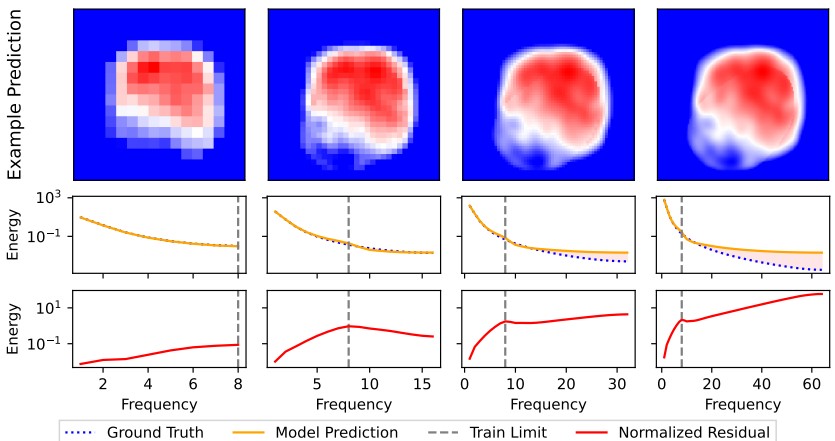

(a) **Zero-Shot Super-Resolution.** Model trained on resolution 16 data, and evaluated at varying resolutions: 16, 32, 64, 128. **Top Row:** Sample prediction for Darcy flow. **Middle Row:** Average test set 2D energy spectrum of label and model prediction. **Bottom Row:** Average residual spectrum normalized by label spectrum.

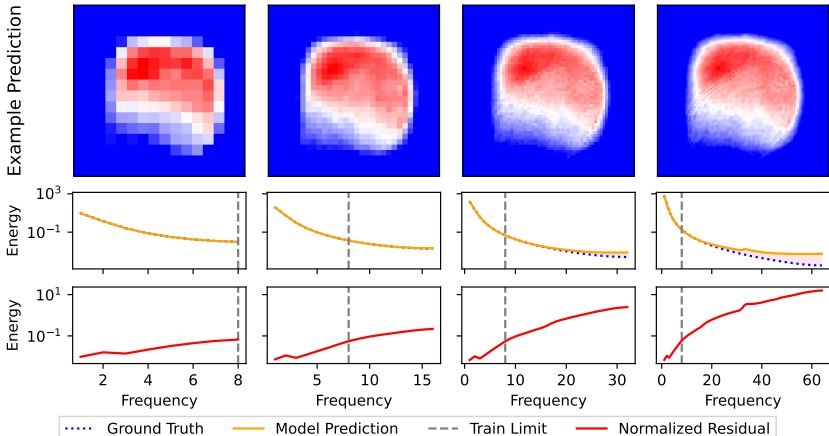

(b) **Multi-resolution Training.** Model trained on dataset with {25% resolution 16, 25% resolution 32, 25% resolution 64, 25% resolution 128} data, and evaluated at varying resolutions: 16, 32, 64, 128. **Top Row:** Sample prediction for Darcy flow. **Middle Row:** Average test set 2D energy spectrum of label and model prediction. **Bottom Row:** Average residual spectrum normalized by label spectrum.

Figure 52: **AMD CPU.** Notice that the model predicted spectra diverge in the zero-shot super-resolution setting after frequency 8 (the maximum frequency present in the training data). Overall, the multi-resolution model predicted spectra are more accurate.

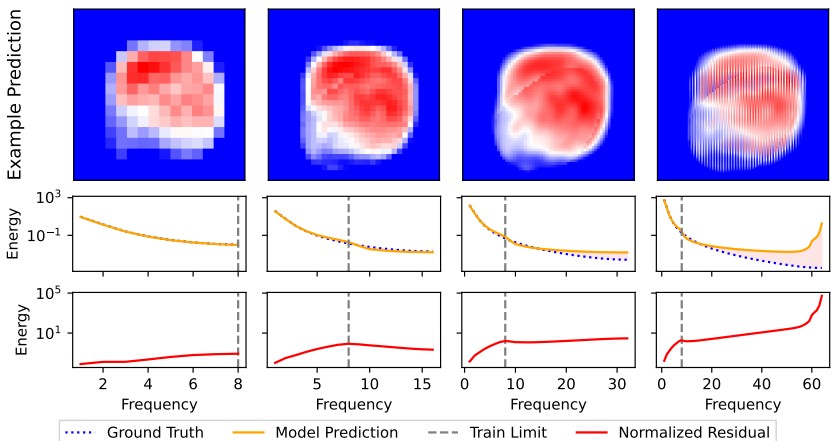

(a) **Zero-Shot Super-Resolution.** Model trained on resolution 16 data, and evaluated at varying resolutions: 16, 32, 64, 128. **Top Row:** Sample prediction for Darcy flow. **Middle Row:** Average test set 2D energy spectrum of label and model prediction. **Bottom Row:** Average residual spectrum normalized by label spectrum.

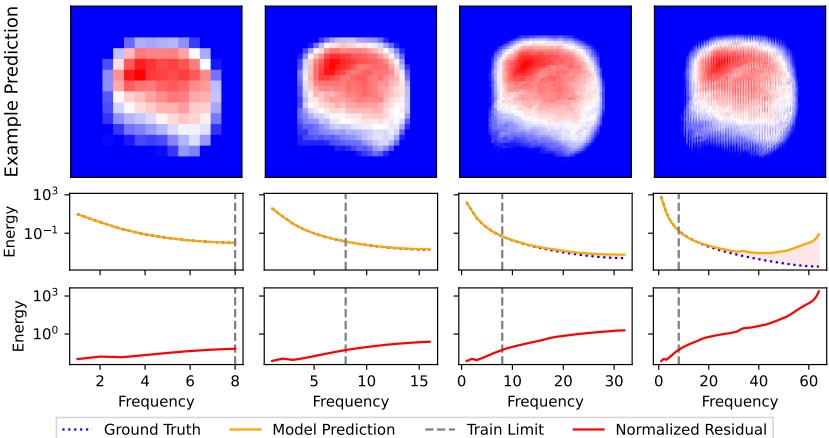

(b) **Multi-resolution Training.** Model trained on dataset with {25% resolution 16, 25% resolution 32, 25% resolution 64, 25% resolution 128} data, and evaluated at varying resolutions: 16, 32, 64, 128. **Top Row:** Sample prediction for Darcy flow. **Middle Row:** Average test set 2D energy spectrum of label and model prediction. **Bottom Row:** Average residual spectrum normalized by label spectrum.

Figure 53: **NVIDIA A100**. Notice that the model predicted spectra diverge in the zero-shot super-resolution setting after frequency 8 (the maximum frequency present in the training data). Overall, the multi-resolution model predicted spectra are more accurate.

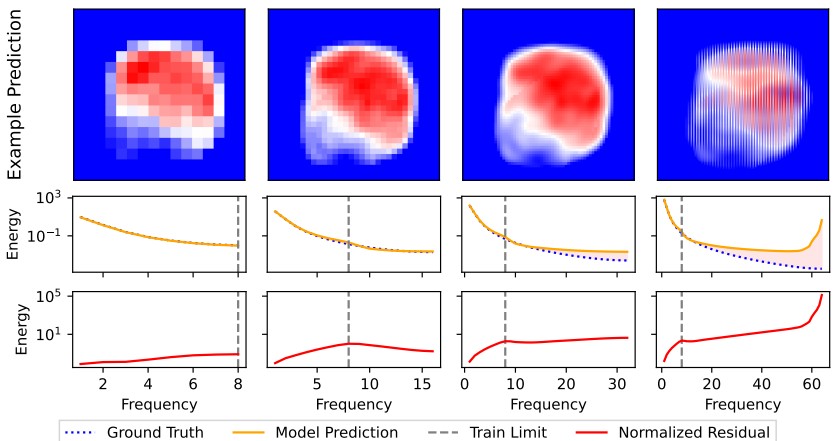

(a) **Zero-Shot Super-Resolution.** Model trained on resolution 16 data, and evaluated at varying resolutions: 16, 32, 64, 128. **Top Row:** Sample prediction for Darcy flow. **Middle Row:** Average test set 2D energy spectrum of label and model prediction. **Bottom Row:** Average residual spectrum normalized by label spectrum.

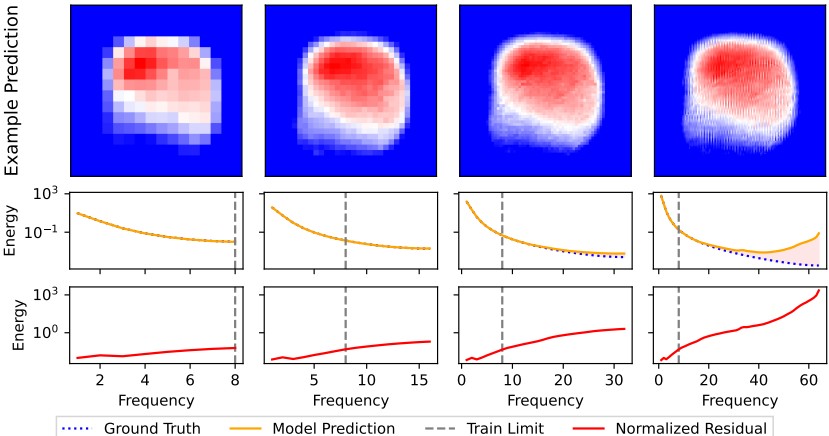

(b) **Multi-resolution Training.** Model trained on dataset with {25% resolution 16, 25% resolution 32, 25% resolution 64, 25% resolution 128} data, and evaluated at varying resolutions: 16, 32, 64, 128. **Top Row:** Sample prediction for Darcy flow. **Middle Row:** Average test set 2D energy spectrum of label and model prediction. **Bottom Row:** Average residual spectrum normalized by label spectrum.

Figure 54: **NVIDIA A40.** Notice that the model predicted spectra diverge in the zero-shot super-resolution setting after frequency 8 (the maximum frequency present in the training data). Overall, the multi-resolution model predicted spectra are more accurate.

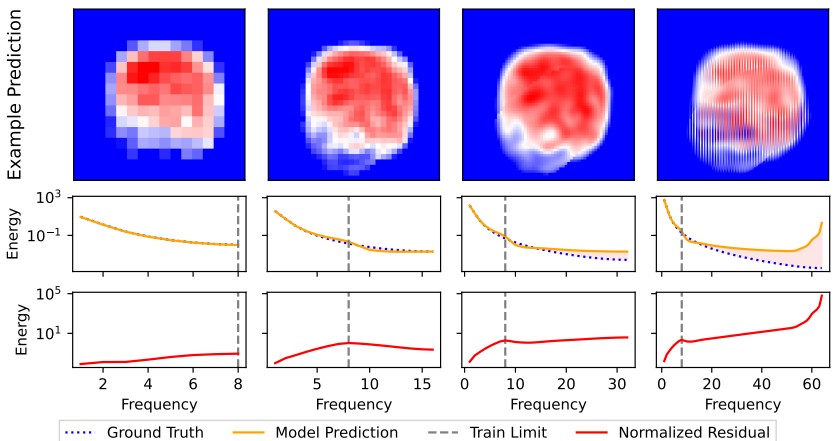

(a) **Zero-Shot Super-Resolution.** Model trained on resolution 16 data, and evaluated at varying resolutions: 16, 32, 64, 128. **Top Row:** Sample prediction for Darcy flow. **Middle Row:** Average test set 2D energy spectrum of label and model prediction. **Bottom Row:** Average residual spectrum normalized by label spectrum.

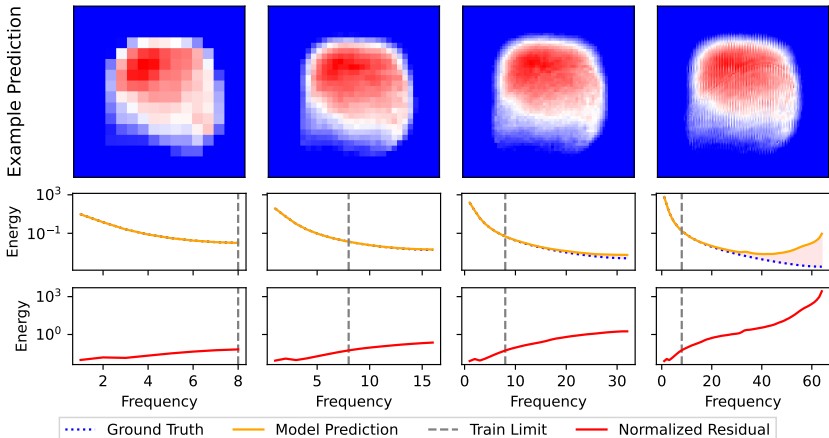

(b) **Multi-resolution Training.** Model trained on dataset with {25% resolution 16, 25% resolution 32, 25% resolution 64, 25% resolution 128} data, and evaluated at varying resolutions: 16, 32, 64, 128. **Top Row:** Sample prediction for Darcy flow. **Middle Row:** Average test set 2D energy spectrum of label and model prediction. **Bottom Row:** Average residual spectrum normalized by label spectrum.

Figure 55: **NVIDIA H100**. Notice that the model predicted spectra diverge in the zero-shot super-resolution setting after frequency 8 (the maximum frequency present in the training data). Overall, the multi-resolution model predicted spectra are more accurate.

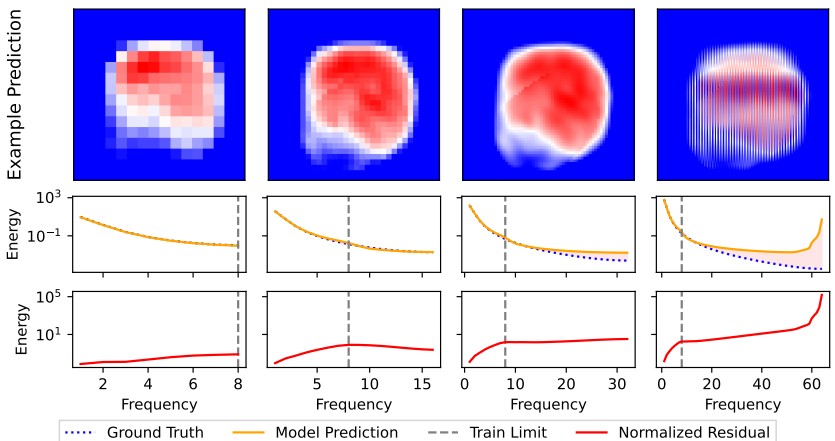

(a) **Zero-Shot Super-Resolution.** Model trained on resolution 16 data, and evaluated at varying resolutions: 16, 32, 64, 128. **Top Row:** Sample prediction for Darcy flow. **Middle Row:** Average test set 2D energy spectrum of label and model prediction. **Bottom Row:** Average residual spectrum normalized by label spectrum.

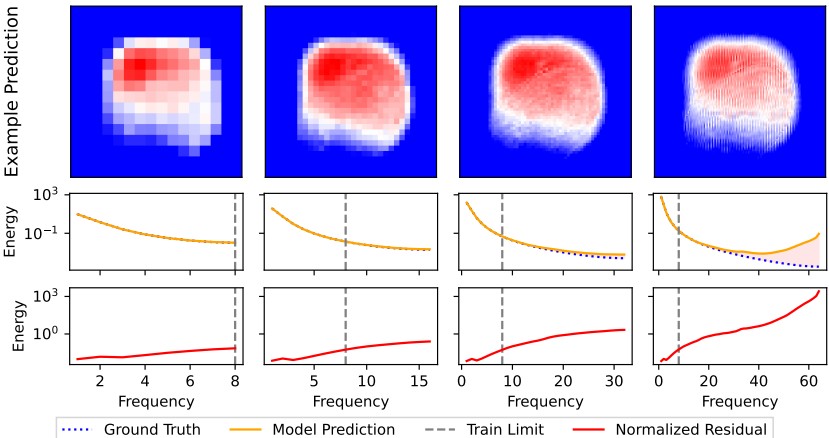

(b) **Multi-resolution Training.** Model trained on dataset with {25% resolution 16, 25% resolution 32, 25% resolution 64, 25% resolution 128} data, and evaluated at varying resolutions: 16, 32, 64, 128. **Top Row:** Sample prediction for Darcy flow. **Middle Row:** Average test set 2D energy spectrum of label and model prediction. **Bottom Row:** Average residual spectrum normalized by label spectrum.

Figure 56: **NVIDIA H200.** Notice that the model predicted spectra diverge in the zero-shot super-resolution setting after frequency 8 (the maximum frequency present in the training data). Overall, the multi-resolution model predicted spectra are more accurate.

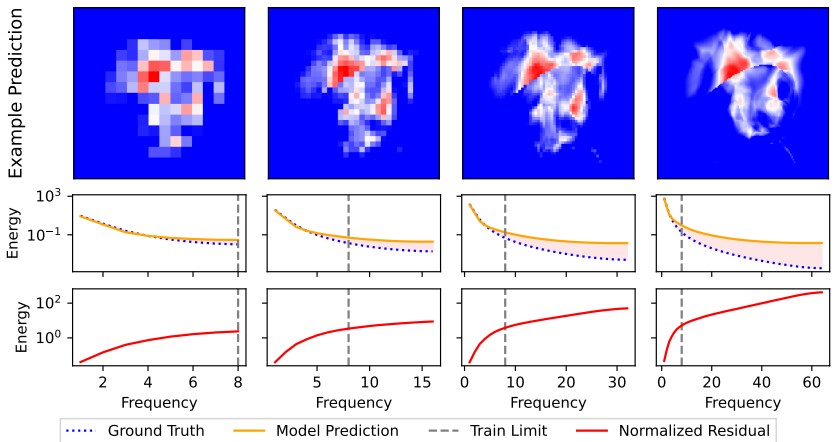

(a) **Zero-Shot Super-Resolution.** Model trained on resolution 16 data, and evaluated at varying resolutions: 16, 32, 64, 128. **Top Row:** Sample prediction for Darcy flow. **Middle Row:** Average test set 2D energy spectrum of label and model prediction. **Bottom Row:** Average residual spectrum normalized by label spectrum.

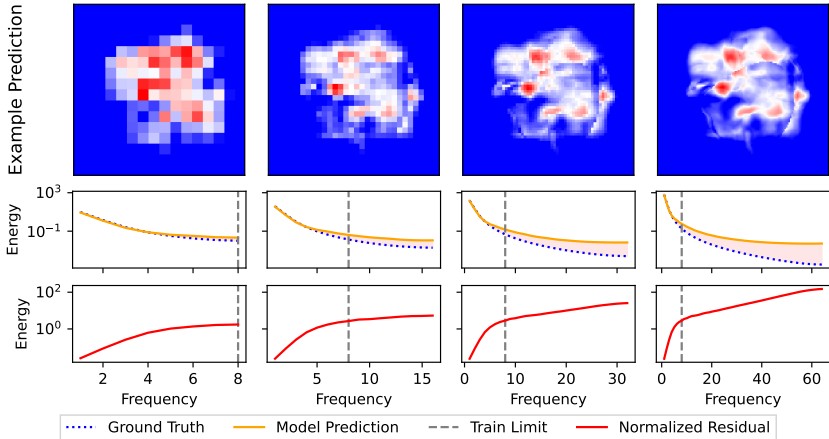

(b) **Multi-resolution Training.** Model trained on dataset with {25% resolution 16, 25% resolution 32, 25% resolution 64, 25% resolution 128} data, and evaluated at varying resolutions: 16, 32, 64, 128. **Top Row:** Sample prediction for Darcy flow. **Middle Row:** Average test set 2D energy spectrum of label and model prediction. **Bottom Row:** Average residual spectrum normalized by label spectrum.

Figure 57: **Apple M2 Pro**. Notice that the model predicted spectra diverge in the zero-shot super-resolution setting after frequency 8 (the maximum frequency present in the training data). Overall, the multi-resolution model predicted spectra are more accurate. **Note:** While the FNO model is capable of inference on this computing architecture, we observe significant distortions in the model predictions; therefore, we suspect the hardware-compatible PyTorch version 2.10.0 for the Apple M2 Pro may not be fully compatible with the "neuraloperator" V2.0 library.

