# OpenReview forum: "The False Promise of Zero-Shot Super-Resolution in Machine-Learned Operators"
_ICLR.cc/2026/Conference — ICLR 2026 Poster_

### Official Review · Reviewer_eXr2 · 2025-10-26

**Soundness:** 3
**Presentation:** 3
**Contribution:** 2
**Rating:** 4
**Confidence:** 3

**Summary:**

The paper evaluates the hypothesis that Fourier Neural Operators (FNOs) and similar Machine-Learned Operators (MLOs) can generalize across resolutions, i.e., perform "zero-shot super-resolution", when trained on low-resolution data. They test the hypothesis on Darcy flow, Burgers', and Navier-Stokes equations and show that FNOs fail to generalize to unseen resolutions or frequency bands. They contribute these failurs as aliasing phenomena and propose a simple fix: multi-resolution training, mixing low- and high-resolution samples, which improves generalization without major cost increase.

**Strengths:**

- The work has clear, well-motivated experiments
- The systematic decomposition of the problem (frequency extrapolation vs resolution interpolation) is nice

**Weaknesses:**

- "zero-shot super-resolution" phrasing misleads readers from computer vision/image SR domains; the task is not SR in the standard sense.
- The insight that trained models don't extrapolate beyond training distribution is general ML knowledge and the work contributes only limited conceptual novelty.
- The proposed solution (multi-resolution training) is expected and straightforward with limited novelty for the ML community.

**Questions:**

The main claims appear to echo the insights of the following work:
[1] Neural Operator: Learning Maps Between Function Spaces With Applications to PDEs, Kovachki et al., JMLR 2023
[2] Discretization-invariance? On the Discretization Mismatch Errors in Neural Operators, Gao et al., ICLR 2025

Could you clearly explain how your analysis provides new insight beyond these works? In particular, is the aliasing perspective quantitatively novel, or primarily a reinterpretation of previously reported resolution-mismatch failures?

---

> ### Author Response · Authors · 2025-11-16
>
> Thank you for your review. We are glad you find our experiments and problem decomposition to be clear, systematic, and well-motivated.
>
> **RE standardization of language/SR discussion:** We acknowledge the definition of super-resolution in a canonical computer vision setting as pertaining to methods that increase the resolution of a given (lower) resolution image [4]. In the scientific ML community however, the “zero-shot super-resolution” terminology was established (to the best of our knowledge) and widely popularized by [3] (e.g., see subsequent works [1, 5, 6, 7]). In [3], **“zero-shot SR”** is defined as a behavior of a model: models trained on low resolution data are able to accurately serve inference on high-resolution data (e.g., in a zero-shot manner). Because of the term’s prominence in the scientific ML community, we choose to not only use the same language, but also to interrogate the claim that zero-shot SR is a performant attribute of machine-learned operators in the first place.
>
> **RE Generalization Beyond Training Data:**  We agree that models do not often generalize beyond their training data. The significance and novelty on this front is the methodology we introduce to assess if data is out-of-distribution w.r.t. the training data. To the best of our knowledge, we are the first to both formalize the decomposition of zero-shot SR into frequency interpolation and resolution interpolation, and then use this decomposition to independently assess MLO’s ability to do both tasks. Further, we emphasize the importance of our study in light of a long line of prior work claiming zero-shot super-resolution to be performant features of certain MLO architectures (predominantly, the FNO architecture) [3]. Our work establishes not only that zero-shot super-resolution is an out-of-distribution task, but provides a simple and actionable data-driven alternative (multi-resolution training) to enable multi-resolution inference. Most importantly, our work is an important stepping stone in establishing an understanding of the limits modeling continuous systems via discrete data.
>
> **RE Simplicity of Multi-Resolution Training:** We are glad that you find our proposed solution to be simple and straightforward. We find the practical nature of our multi-resolution training strategy ensures that it is both **actionable and adoptable** in existing scientific machine learning workflows (unlike more complicated alternatives). Despite the simplicity of multi-resolution training, it results in significant computational performance benefits; in addition to the established multi-resolution modeling benefits (Figure 9). We include an **additional** quantitative analysis of both the reduction in dataset size/memory requirements and reduction in training wall-clock time in Appendix I.1. For Darcy, Burgers, and Navier-Stokes we **document a 96%, 98%, and 96% decrease in dataset size/memory requirements** when comparing multi-resolution training to the highest-resolution single-resolution training; this is a significant decrease in dataset size which results in notable reduction in training time documented in Table 4.
>
> **In response to all reviewers, we have uploaded an updated PDF of our paper with an updated appendix. We summarize the updates here:**
> - Appendix B: Model configuration details
> - Appendix D.1: Clarification of our Filtering/Downsampling procedures
> - Reproducibility statement
> - Appendix J: Experiments w/ DeepONet
> - Appendix K: Experiments w/ Anti-aliasing nonlinearities
> - Appendix L: Multi-seeded experiments and sensitivity analysis
> - Appendix I.1: Computational measurements of training time/memory savings from multi-res training

---

> > ### Author Response · Authors · 2025-11-16
> >
> > **RE Discussing Prior Literature:** We cite both [1] and [2] in our paper. To expand:
> > - [1] defines “discretization-invariant” models as models which can: 1) act on any input discretization, 2) be evaluated at any point in the output domain and 3) converge to a continuum operator as the discretization is refined (section 1.1). It then makes the claim that discretization-invariant models can do zero-shot super-resolution: “The operator setting leads to an efficient and accurate representation, and the resolution-invariant properties make it possible to training and a smaller resolution dataset, and be evaluated on arbitrarily large resolution” and “Furthermore, resolution-invariant operators can do zero-shot super-resolution, as shown in Subsubection 7.3.1”. **Our work directly interrogates the claims made in [1], and shows machine-learned operators are not able to do zero-shot SR via effective problem decomposition in the spectral domain (resolution interpolation & information extrapolation).**
> > - [2] first empirically shows that machine learned operators struggle to do zero-shot multi-resolution inference via L2 error and then proposes CROP to overcome this limitation. We systematically expand their initial study by introducing a **comprehensive spectral analysis for why** MLOs struggle to do zero-shot SR (e.g., resolution interpolation & information extrapolation). We then use these metrics to assess [2]’s proposed solution. [2] proposes a band-limited learning method, Cross-Resolution Operator Learning Pipeline (CROP), to attempt to enable effective multi-resolution inference. We use CROP as a baseline method in our study (section 4.2); we find that while CROP effectively fits the low frequencies, it, by design, fails to fit the higher-frequencies in data. Thus we conclude that the band-limited nature of CROP is not ideal for accurate multi-resolution inference in which a wide range of frequencies must be accurately modeled. To overcome this limitation, we propose data-driven multi-resolution training.
> >
> > [1] Neural Operator: Learning Maps Between Function Spaces With Applications to PDEs, Kovachki et al., JMLR 2023.
> >
> > [2] Discretization-invariance? On the Discretization Mismatch Errors in Neural Operators, Gao et al., ICLR 2025
> >
> > [3] Fourier neural operator for parametric partial differential equations, ICLR, 2021.
> >
> > [4] Learning a deep convolutional network for image super-resolution, ECCV, 2014.
> >
> > [5] Vectorized Conditional Neural Fields: A Framework for Solving Time-dependent Parametric Partial Differential Equations, ICML, 2024.
> >
> > [6] Zero-Shot Super-Resolution from Unstructured Data Using a Transformer-Based Neural Operator for Urban
> > Micrometeorology, 2025.
> >
> > [7] Incremental Spatial and Spectral Learning of Neural Operators for Solving Large-Scale PDEs, TMLR, 2024.

---

> ### Author Response · Authors · 2025-11-24
>
> We would like to check in to see if you found our additional paper contributions useful to address your comments. If so, can you update your score accordingly? If not can you please be more specific as to what may be helpful for us to address during the rebuttal period? We would be glad to clarify any additional details.

---

### Official Review · Reviewer_2rDL · 2025-10-31

**Soundness:** 3
**Presentation:** 3
**Contribution:** 3
**Rating:** 6
**Confidence:** 1

**Summary:**

The paper validates the ability of trained MLOs to generalize beyond their training resolution and shows that accurate zero-shot multi-resolution inference is unreliable.

They find that neither approach, incorporating physics-informed constraints during
training and performing band-limited learning, enables reliable multi-resolution generalization.

Also, the authors propose and test multi-resolution training, where they include training data of varying
resolutions (in particular, a small amount of expensive higher-resolution data and a larger
amount of cheaper lower-resolution data).

**Strengths:**

1. They systematically analyze the zero-shot multi-resolution abilities of FNO (Fourier Neural Operator) with several methods including resolution interpolation and information extrapolation

2. The paper visualizes the results and their findings clearly which leads to better readability of the paper.

**Weaknesses:**

1. The paper does not show how the findings in the paper can be applied to real applications including deep learning-based training/inference.

**Questions:**

1. How the findings in the paper can be applied to the real-world problem?

---

> ### Author Response · Authors · 2025-11-16
>
> Thank you for the review. We are glad you find our study to be both systematic from an experimental perspective and clear.
>
> **RE Real-world problem:** Modeling scientific phenomena is critical to many science and engineering applications. The scientific datasets used in our paper (Darcy Flow, Burgers, and Navier-Stokes) are all well studied fluid flow datasets. The efficient and accurate modeling of fluid flow is critical to applications like modeling blood flow within the human body, ocean currents, and how air flows around airplanes to list a few. An important aspect of these modeling problems is that they can be represented at varying resolutions. For example you can model fluid flow on the scale of square meters or square kilometers; the key difference in both settings is the level of detail which will be modeled. In safely critical settings, like modeling turbulent fluid flow around an airplane, it is very important that the fine-grained details are accurately represented in the modeled simulation. The ultimate goal of our study is to assess if prior work can accurately do multi-resolution inference in a zero-shot manner (we find it cannot), and then we propose a simple, computationally inexpensive solution which reliably enables accurate multi-resolution modeling.
>
> **RE Application to deep-learning based training/inference:** Our study critically examines a popular deep-learning-based class of models (Machine Learned Operators, MLOs) which claim to accurately model these fluid flows at varying resolutions. The critical claim in prior work is that models are able to train on low-resolution data and then do accurate inference on high-resolution data (e.g., zero-shot super-resolution) [1]. To the best of our knowledge, we are the first to both formalize the decomposition of zero-shot super-resolution into frequency interpolation and resolution interpolation, and then use this decomposition to independently assess MLO’s ability to do both tasks. Further, we emphasize the importance of our study in light of a long line of prior work claiming zero-shot super-resolution to be performant features of certain MLO architectures (predominantly, the FNO architecture) [1]. Our work establishes not only that zero-shot super-resolution is an out-of-distribution task, but provides a simple and actionable data-driven alternative (multi-resolution training) to enable multi-resolution inference.
>
> **In response to all reviewers, we have uploaded an updated PDF of our paper with an updated appendix. We summarize the updates here:**
> - Appendix B: Model configuration details
> - Appendix D.1: Clarification of our Filtering/Downsampling procedures
> - Reproducibility statement
> - Appendix J: Experiments w/ DeepONet
> - Appendix K: Experiments w/ Anti-aliasing nonlinearities
> - Appendix L: Multi-seeded experiments and sensitivity analysis
> - Appendix I.1: Computational measurements of training time/memory savings from multi-res training
>
>
> [1] Fourier neural operator for parametric partial differential equations, ICLR, 2021.

---

> > ### Author Response · Authors · 2025-11-24
> >
> > We would like to check in to see if you found our additional paper contributions useful to address your comments. If so, can you update your score accordingly? If not can you please be more specific as to what may be helpful for us to address during the rebuttal period? We would be glad to clarify any additional details.

---

### Official Review · Reviewer_MdwW · 2025-10-31

**Soundness:** 3
**Presentation:** 4
**Contribution:** 3
**Rating:** 6
**Confidence:** 4

**Summary:**

The paper interrogates the widely circulated claim that mesh‑invariant machine‑learned operators (MLOs)—notably Fourier Neural Operators (FNOs)—can do zero‑shot multi‑resolution inference: train on one grid and generalize to higher (or lower) resolutions without additional data. It decomposes the problem into two behaviors: resolution interpolation (same frequency content, new sampling rate) and information extrapolation (same sampling rate, new frequency content). Carefully controlled experiments on Darcy (2D), Burgers (1D), and incompressible Navier–Stokes (2D) from PDEBench reveal that FNOs fail at both: residual energy spectra spike exactly in bands that should remain quiet under the experimental controls, evidencing aliasing; artifacts grow through time‑rollouts for Navier–Stokes. “Zero‑shot fixes”—adding physics‑informed losses or adopting band‑limited approaches (CNO, CROP)—do not rescue multi‑resolution generalization: physics terms usually hurt, and band‑limited models fit low frequencies but drop high‑frequency content by design. The authors then propose a simple multi‑resolution training recipe (mix mostly low‑res with a small fraction of high‑res), documenting a favorable cost–accuracy Pareto front across tasks. Figures and appendices provide spectral diagnostics, max‑modes ablations, and resolution×resolution heatmaps that collectively undermine the zero‑shot super‑resolution narrative and replace it with an actionable practice.

**Strengths:**

Good conceptual framing: Separating resolution interpolation from information extrapolation clarifies why “train low, test high” fails and grounds the evaluation in sampling theory.

Rigorous diagnostics: Normalized residual energy spectra consistently reveal aliasing when resolution or frequency content shift, time‑rollout visuals show error compounding in Navier–Stokes.

Breadth of evaluation: Three canonical PDEs, systematic train/test grids, and resolution×resolution loss heatmaps; max‑modes ablations show failures persist across spectral truncation choices.

Actionable guidance: Multi‑resolution training (mostly low‑res + some high‑res) improves robustness with modest cost, forming a clean Pareto front between average data size and error.

The study complements claims of discretization invariance in operator learning (FNO, DeepONet, mesh‑invariant variants) and recent aliasing/anti‑aliasing discussions by providing decisive, controlled counter‑evidence and a practical alternative.

**Weaknesses:**

1. Most conclusions rest on FNO (plus CNO/CROP wrappers). Including non‑Fourier operator learners (e.g., DeepONet/U‑NO/multiwavelet or graph‑kernel operators) would better support claims that span “MLOs” broadly.

2. Results appear single‑seed without confidence intervals or standard deviation reporting, negative conclusions look stable but would benefit from variance reporting across seeds and data resamplings.

3. In turbulence, band‑limited methods can achieve competitive MSE while failing spectrally; adding physics‑aware diagnostics (e.g., enstrophy/energy spectra errors, divergence) would clarify practical significance.

4. While max‑modes are ablated, there is no explicit test of antialiased nonlinearity/resize operations (well‑known in vision/GANs) inside FNO blocks; this could disentangle architecture‑ vs. data‑pipeline‑driven aliasing.

5. The study focuses on resolution/frequency shifts; it would be useful to probe parameter OOD (e.g., viscosity/forcing) to test whether the proposed multi‑resolution recipe transfers to broader distribution shifts.

6. There is a qualitative data‑size vs. epoch‑time trend, but a more explicit accounting (wall‑clock, memory, FLOPs) for the zero‑shot baselines vs. multi‑resolution training would sharpen the practical trade‑offs.

**Questions:**

1. Evaluation is limited primarily to Fourier-based operator architectures. Including at least one non-Fourier, mesh-invariant operator (e.g., DeepONet, U-NO, or multiwavelet neural operator) would strengthen the generality of the findings and demonstrate whether the observed zero-shot failure modes persist across different operator families.

2.  For the Navier–Stokes experiments, additional quantitative diagnostics—such as errors on enstrophy, energy spectra, or divergence and conservation measures—are needed to clarify the apparent discrepancy between low mean-squared error and degraded spectral fidelity in band-limited models.

3. An explicit investigation of anti-alias filtering around nonlinearities, or of antialiased up-/down-sampling within the operator pipeline, can determine whether the reported interpolation and extrapolation failures are due to architectural aliasing rather than inherent data-distribution shifts.

4. The study shows that physics-informed losses underperform across weighting choices. Testing alternative formulations—such as residual-only constraints, adaptive weighting or curriculum schemes, or discretization-matched residuals—could be informative to rule out optimization artifacts and isolate the underlying cause of this degradation.

5.  Reporting mean ± standard deviation across multiple random seeds for key figures and tables, together with an ablation of the low-/high-resolution data ratios in the multi-resolution training protocol, are needed for more robust evidence and practical guidance for determining data-budget trade-offs.

6. It remains unclear whether the proposed multi-resolution training improves robustness to out-of-distribution conditions beyond discretization changes, such as shifts in physical parameters (e.g., viscosity, forcing, boundary conditions).

7.  Public release of scripts for generating controlled low-pass and resampled datasets, computing normalized residual spectra, and reproducing the reported Pareto-front trade-off plots are required for areproducibility and transparency.

---

> ### Author Response · Authors · 2025-11-16
>
> Thank you for your review. We are glad you found our work to have a good conceptual framing and that it provides decisive, controlled counter‑evidence and a practical alternative (multi-resolution training).
>
> In response to your queries, we **update** our paper with additional details:
> - **RE Additional (non Fourier-based) Machine-learning Operators:** We focus our initial investigation on FNOs as they are the first prominent MLO to claim zero-shot super-resolution. Following our investigation of FNOs we then study 3 additional MLO baselines:
>   - Physics-informed Neural Operator (PINO, Section 4.1)
>   - Convolutional Neural Operator (CNO, Section 4.2)
>   - Cross-Resolution Operator Learning (CROP, Section 4.2)
>
>   **We note that the CNO is a non-Fourier based model.** To expand the breadth of our analysis, we also include an **additional** analysis of DeepONet in **Appendix J**. While DeepONet does not make the claim of **zero-shot** super-resolution, we assess its ability to do resolution interpolation, information extrapolation, and zero-shot super-/sub-resolution abilities. We find that DeepONet fails to achieve all three objectives (Figures 34-37). We then assess if multi-resolution training can enable data-driven multi-resolution inference with DeepONet and again confirm that the model is most performant on the data resolutions seen during training (Figure 38). Our findings for DeepONet are consistent with the trends we observe for FNO, as expected.
> - **RE Public Release of Scripts and Reproducibility Material:** We include an updated reproducibility statement which references details  about hyperparameter tuning in Appendix A, model details in Appendix B, public datasets configuration in Appendix C, data extrapolation/interpolation implementation details in Appendix D.1. To ensure reliability of results we include an additional sensitivity analysis in Appendix L. To respect double-blind review, we initially omit including a link to all of our experimental code; this will be released after the review period.
> - **RE Assessing Impact of Anti-Aliasing Activation Functions:** First, we note that the CNO architecture (which is already included in our analysis) uses an anti-aliasing activation function, and we have shown in Section 4.2 that CNO cannot accurately do zero-shot super-/sub-resolution. **To further explore this impact, we include a new analysis of the impact of anti-alias nonlinearities to disentangle architecture- vs. data-pipeline-driven aliasing in Appendix K.** We repeat the resolution interpolation, information extrapolation, and zero-shot super-resolution and sub-resolution experiments for FNO w/ anti-aliasing activation function and find that aliasing is still present in all experiments. This confirms that the out-of-distribution nature of data plays a significant role in aliasing for MLOs.
> - **RE Multiple Seeds/Sensitivity Analysis:** Due to the computational cost of experiments, we initially ran a single seed per experiment. We ensured the robustness of our results by replicating observed trends across a wide array of datasets (Darcy, Burgers, Navier-Stokes) as a proxy for generalizability of results. Despite this we now include **additional** sensitivity analysis via multi-seeded experiments in Appendix L. We repeat the resolution interpolation, information extrapolation, and **zero-shot** super- and sub-resolution experiments for Darcy Flow across 3 random seeds; as expected, the results are consistent with what was already observed. We report average **zero-shot** super-/sub-resolution performance of models with standard deviations in Table 3. An ablation of the low-/high-resolution data ratios in the multi-resolution training protocol are already included for all datasets (Darcy, Burgers, Navier-Stokes) across multiple ratios (0.05, 0.1, 0.25, 0.5, 0.75, 0.9, 0.95) in **Appendix I**. Observed trends are consistent across ablations; models perform best on training resolutions seen during training.
> - **RE Wall-Clock Time/Memory Savings from Multi-Res training:** We documented reduction in dataset size from multi-resolution training in Figure 9. We **additionally** include a new quantitative analysis of both the reduction in dataset size/memory requirements and reduction in training wall-clock time in **Appendix I.1**. For Darcy, Burgers, and Navier-Stokes we document a **96%, 98%, and 96% decrease in dataset size/memory requirements** when comparing multi-resolution training to the highest-resolution single-resolution training; this is a significant decrease in dataset size and results in notable reduction in training time documented in table 4.

---

> > ### Author Response · Authors · 2025-11-16
> > **Responding to additional questions**
> >
> > - **RE Extended Finding to Other Forms of OOD Data:** OOD generalization in SciML has multiple axes (resolution, PDE parameters, forcing/boundary conditions, and geometry) each introducing different types of unseen information. Our work focuses on the resolution axis, which is intrinsic to super-resolution: changing resolution changes the frequency content of a data sample. Other OOD shifts, such as parameter changes, also change the frequency content, but in a fundamentally different way: they alter the underlying dynamics rather than the sampling of the same dynamics. While additional data can help within a given parameter regime, multi-resolution training on one parameter range would not be expected to generalize to a disjoint parameter range, because the added information is qualitatively different. Therefore, developing efficient data sampling schemes with the goal of enabling cross-parameter generalization, similar to our multi-resolution training scheme which enables cross-resolution generalization, remains an exciting and viable future direction.
> > - **RE Physics-informed constraints:** We agree that there are likely several optimization-based reasons for why physics-informed constraints underperform their data-driven counterparts. This is a well known result [1] and we discuss this in our paper (lines 306-308). The objective of our study is to provide fair comparisons between methods; because physics-informed optimization is inherently challenging, adding additional axes such as residual-only constraints, adaptive weighting or curriculum schemes, or discretization-matched residuals, would simply increase the computational cost and complexity of the physics-informed solution compared to the other studied methods. Instead, proposed solutions like multi-resolution training do not require such tuning and remain performant for a fraction of the computation cost.
> > - **RE Navier-Stokes Diagnostics:** We already include the energy spectra for the Navier-Stokes experiments for the band-limited setting in Figure 24; we juxtapose the spectral error with the MSE error. We discuss in Appendix H that the long-tailed energy distribution across frequency modes is what results in the low MSE and correspondingly **masks the large discrepancy** between the prediction and ground truth for the high-frequency modes; in settings which modeling high-frequency modes is critical (e.g., turbulence), this is a notable error.
> >
> > **In response to all reviewers, we have uploaded an updated PDF of our paper with an updated appendix. We summarize the updates here:**
> > - Appendix B: Model configuration details
> > - Appendix D.1: Clarification of our Filtering/Downsampling procedures
> > - Reproducibility statement
> > - Appendix J: Experiments w/ DeepONet
> > - Appendix K: Experiments w/ Anti-aliasing nonlinearities
> > - Appendix L: Multi-seeded experiments and sensitivity analysis
> > - Appendix I.1: Computational measurements of training time/memory savings from multi-res training
> >
> > [1] Characterizing possible failure modes in physics-informed neural networks. NEURIPS, 2021.

---

> > > ### Comment · Reviewer_MdwW · 2025-11-21
> > >
> > > I thank the authors for additional experiments and details included in the revised manuscript. Based on the authors’ detailed response and the evaluations provided by the other reviewers, I would keep my rating at this time, however, the authors are encouraged to address the following additional point.
> > >
> > > RE Public Release of Scripts and Reproducibility Material: The authors should include an anonymous link to the codebase - this will ensure double blind policy is respected while making the code available to the community. https://anonymous.4open.science/

---

> > > > ### Author Response · Authors · 2025-11-22
> > > >
> > > > Thank you for sharing that anonymization resource! We have uploaded a link to our anonymized code in the supplementary materials.
> > > >
> > > > Please let us know if you have any additional questions or if we have satisfied all of your concerns.

---

### Official Review · Reviewer_am35 · 2025-11-02

**Soundness:** 3
**Presentation:** 3
**Contribution:** 4
**Rating:** 8
**Confidence:** 3

**Summary:**

This paper empirically investigates whether Machine-Learned Operators (MLOs) can reliably generalize across different resolutions when modeling Partial Differential Equations (PDEs), a capability often referred to as zero-shot super-resolution or sub-resolution. Through well-structured experiments covering interpolation, extrapolation, and super-resolution scenarios, the authors demonstrate that Fourier Neural Operators (FNOs) consistently fail when evaluated at unseen resolutions, largely due to aliasing and out-of-distribution effects. The study further shows that physics-constrained optimization and band-limited training, do not address these issues. As a mitigation strategy, the authors propose a simple multi-resolution training protocol, which substantially improves accurate results but still falls short under severe resolution mismatches.

**Strengths:**

It is clearly written, well-structured, and addresses an important yet often overlooked question in the MLOs. The authors effectively decompose the problem into resolution interpolation, information extrapolation, and both, providing a rigorous framework for analysis. The work’s key contribution lies in extensive empirical evaluation under different settings.

**Weaknesses:**

While the title refers broadly to “Machine-Learned Operators (MLOs),” the experiments primarily focus on the FNOs. Given that FNOs represent only one class of MLO architectures, extending the analysis or discussion to other operator-learning models would help clarify how general these findings truly are.

**Questions:**

- A single consolidated table summarizing final model configurations (e.g., number of modes, hidden dimensions, etc.) for each dataset would improve clarity and ease of implementation.

- Since aliasing is highly sensitive to data resampling, the paper should more clearly specify the interpolation and filtering operations used in generating different resolutions.

- While the empirical evidence is compelling, the paper would benefit from a brief theoretical discussion.

---

> ### Author Response · Authors · 2025-11-16
>
> Thank you for your review. We are glad you found our work to both decompose the problem effectively and provide an extensive empirical evaluation.
>
> In response to your queries, we **update** our paper with additional details:
> - **RE Consolidation of Model Configurations:** Please find all model configuration details and links to their original implementation in **Appendix B**.
> - **RE Specification of interpolation and filtering operations:** Please find all details w.r.t. how data were downsampled and filtered in **Appendix D.1**.
> - **RE Additional Machine-learning Operators:** We focus our initial investigation on FNOs as they are the first prominent MLO to claim zero-shot super-resolution. Following our investigation of FNOs we then study 3 additional MLO baselines:
>   - Physics-informed Neural Operator (PINO, Section 4.1)
>   - Convolutional Neural Operator (CNO, Section 4.2)
>   - Cross-Resolution Operator Learning (CROP, Section 4.2)
>
>   To expand the breadth of our analysis, we also include an **additional** analysis of DeepONet in **Appendix J**. While DeepONet does not make the claim of **zero-shot** super-resolution, we assess its ability to do resolution interpolation, information extrapolation, and zero-shot super-/sub-resolution abilities. We find that DeepONet fails to achieve all three objectives (Figures 34-37). We then assess if multi-resolution training can enable data-driven multi-resolution inference with DeepONet and again confirm that the model is most performant on the data resolutions seen during training (Figure 38). Our findings for DeepONet are consistent with the trends we observe for FNO, as expected.
>
> **In response to all reviewers, we have uploaded an updated PDF of our paper with an updated appendix. We summarize the updates here:**
> - Appendix B: Model configuration details
> - Appendix D.1: Clarification of our Filtering/Downsampling procedures
> - Reproducibility statement
> - Appendix J: Experiments w/ DeepONet
> - Appendix K: Experiments w/ Anti-aliasing nonlinearities
> - Appendix L: Multi-seeded experiments and sensitivity analysis
> - Appendix I.1: Computational measurements of training time/memory savings from multi-res training

---

### Author Response · Authors · 2025-12-01
**Summarizing Rebuttal/General Remarks**

Dear AC, SAC, and PCs,

We understand that we are in a difficult period, and the recent unexpected changes have significantly increased your workload. We would like to highlight that each reviewer has noted the work as an important contribution to understanding the limits of scientific machine learning with respect to modeling continuous problems discretely and has acknowledged the simplicity and utility of our proposed solution.

We provide a concise summary of each reviewer’s feedback and our responses below:

**Reviewer MdwW**: Found our work to have a “good conceptual framing” and that it provides “decisive, controlled counter‑evidence and a practical alternative [multi-resolution training]”. This was the only reviewer that was able to respond to our rebuttal; and we were able to address all of their comments.

**Reviewer am35, Reviewer 2rDL, Reviewer eXr2** were unfortunately not able to respond during the rebuttal period, but each found our work to address important questions, effectively decompose the problem, and propose practical solutions to enable multi-resolution scientific modeling.

For example, **Reviewer am35** found our work to address “an important yet often overlooked question in the MLOs”. They subsequently write: “The authors effectively decompose the problem [zero-shot super-resolution] into resolution interpolation, information extrapolation, and both, providing a rigorous framework for analysis”. **Reviewer eXr2**: Found our work to have “clear, well-motivated experiments”, provide a “systematic decomposition of the problem”, and propose a "straightforward" solution.

To help facilitate the decision process, we outline the changes to our revised manuscript here:

- **Appendix B**: Model configuration details
- **Appendix D.1**: Clarification of our Filtering/Downsampling procedures
- **Updated Reproducibility statement + Anonymous Code release**
- **Appendix J**: Experiments w/ DeepONet; we extend our findings to an additional architecture. (We find that DeepONet fails to do zero-shot super-resolution objectives and confirm that multi-resolution training can enable data-driven multi-resolution inference. Our findings for DeepONet are consistent with the trends we observe for Fourier-based MLOs, as expected).
- **Appendix K**: Experiments w/ Anti-aliasing nonlinearities. (We confirm that out-of-distribution nature of data plays a significant role in aliasing for MLO by controlling for the role of aliasing in activation functions; we find that aliasing is still present in all experiments.)
- **Appendix L**: Multi-seeded experiments and sensitivity analysis
- **Appendix I.1**: Computational measurements of training time/memory savings from multi-res training. (We document a 96-98% decrease in dataset size/memory needs for multi-resolution training to the highest-resolution single-resolution training; this is a significant decrease and results in notable reduction in training time.)

We believe the paper has been substantially strengthened through the rebuttal and meets the quality and significance expected for ICLR. Thank you again for your time and consideration.

Best regards,

Authors

---

### Meta-Review · Area_Chair_5FWc · 2026-01-02

**Summary:**

This paper introduces a simple and principled approach to enable accurate multi-resolution inference. The major concerns of this paper include the generalization ability of the proposed method and robustness to out-of-distribution conditions beyond discretization changes.

**Reviewer Concerns:**

As this paper aim to develop a simple, computationally-efficient, and data-driven multi-resolution training protocol, the generalization ability of the proposed method and robustness to out-of-distribution conditions beyond discretization changes should be addressed. In the provided rebuttal, the authors solve these concerns well.

**Reviewer Scores:**

In the provided rebuttal, the authors solve the concerns of reviewers well.

---

### Decision · Program_Chairs · 2026-01-26

Accept (Poster)